

# Long-term global measurements of methanol, ethene, ethyne, and HCN from the Cross-track Infrared Sounder

Kelley C. Wells[1], Dylan B. Millet[1], Jared F. Brewer[1], Vivienne H. Payne[2], Karen E. Cady-Pereira[3], Rick Pernak[3], Susan Kulawik[2,4], Corinne Vigouroux[5], Nicholas Jones[6], Emmanuel Mahieu[7], Maria Makarova[8], Tomoo Nagahama[9], Ivan Ortega[10], Mathias Palm[11], Kimberly Strong[12], Matthias Schneider[13], Dan Smale[14], Ralf Sussmann[15], and Minqiang Zhou[16]

[1]Department of Soil, Water, and Climate, University of Minnesota, Saint Paul, MN, USA
[2]Jet Propulsion Laboratory, California Institute of Technology, Pasadena, CA, USA
[3]Atmospheric and Environmental Research, Inc., Lexington, MA, USA
[4]BAER Institute, Petaluma, CA, USA
[5]Royal Belgian Institute for Space Aeronomy (BIRA-IASB), Brussels, Belgium
[6]School of Chemistry, University of Wollongong, Wollongong, Australia
[7]Institute of Astrophysics and Geophysics, University of Liège, Liège, Belgium
[8]Atmospheric Physics Department, Saint Petersburg State University, Saint Petersburg, Russia
[9]Institute for Space and Earth Environmental Research, Nagoya University, Nagoya, Japan
[10]Atmospheric Chemistry Observations & Modeling, National Center for Atmospheric Research, Boulder, CO, USA
[11]Institute of Environmental Physics, University of Bremen, Bremen, Germany
[12]Department of Physics, University of Toronto, Toronto, ON, Canada
[13]Institute of Meteorology and Climate Research (IMK-ASF), Karlsruhe Institute of Technology, Karlsruhe, Germany
[14]National Institute of Water and Atmospheric Research, Lauder, New Zealand
[15]Karlsruhe Institute of Technology (KIT), IMK-IFU, Garmisch-Partenkirchen, Germany
[16]Institute of Atmospheric Physics, Chinese Academy of Sciences, Beijing, China

*Correspondence to*: Dylan B. Millet (dbm@umn.edu)

**Abstract.** Volatile organic compounds (VOCs) play an important role in modulating the atmosphere's oxidizing capacity and affect tropospheric ozone, carbon monoxide, formaldehyde, and organic aerosol formation. Space-based observations can provide powerful global information to advance our knowledge of these processes and their changes over time. We present here the development of new retrievals for four key VOCs (methanol, ethene, ethyne, and HCN) based on thermal infrared radiance observations from the satellite-borne Cross-track Infrared Sounder (CrIS). We update the Retrieval of Organics from CrIS Radiances (ROCR) algorithm developed previously for isoprene to explicitly account for the spectral signal dependence on the VOC vertical profile shape, and apply this updated retrieval (ROCRv2) to derive column abundances for the targeted species across the full Suomi-NPP CrIS record (2012-2023). The CrIS data are well-correlated with ground-based Network for the Detection of Atmospheric Composition Change (NDACC) retrievals for methanol (r=0.77-0.84); HCN and ethyne exhibit lower correlations (r=0.36-0.44 and 0.56-0.65, respectively) with an apparent 40% CrIS/NDACC disparity for ethyne. The results reveal robust global distributions of the target VOCs from known biogenic, biomass burning, and industrial source regions, and demonstrate the impact of anomalous events such as the 2015-2016 El Niño. They also highlight the importance of accurate vertical profile constraints when evaluating and interpretating thermal infrared data records. Initial comparisons of



the CrIS observations to predicted VOC distributions from the GEOS-Chem chemical transport model point to large uncertainties in our current understanding of the atmospheric ethene budget as well as to underestimated HCN, ethyne, and methanol sources.

## 1 Introduction

Volatile organic compounds (VOCs) play an important role in modulating the atmosphere's oxidizing capacity and in the formation of tropospheric ozone, carbon monoxide (CO), formaldehyde, and organic aerosols. They originate from diverse sources, including biogenic emissions, urban/industrial activities, wildfires, and atmospheric photochemistry; however, large uncertainties in these sources—and the underlying processes that control them—prevent accurate assessment of VOC impacts and their changes over time. Sparse observational coverage for many VOCs has hampered our ability to reduce those uncertainties and advance predictive models. In this paper we present the development and evaluation of new satellite-based measurements for four key atmospheric VOCs: methanol ($CH_3OH$), ethene ($C_2H_4$), ethyne ($C_2H_2$), and hydrogen cyanide (HCN).

Methanol is the most abundant VOC in the atmosphere and an important precursor of ozone, formaldehyde, and carbon monoxide (Tie et al., 2003; Duncan et al., 2007; Hu et al., 2011; Wells et al., 2014). It is emitted primarily from terrestrial plants (Macdonald and Fall, 1993; Galbally and Kirstine, 2002; Warneke et al., 1999), with smaller sources from biomass burning (Holzinger et al., 1999), oceans (Millet et al., 2008), industrial activities (De Gouw et al., 2005), and peroxy radical reactions (Tyndall et al., 2001; Bates et al., 2021). Previous work has identified large seasonal and spatial biases in bottom-up terrestrial methanol emission estimates (Wells et al., 2012; Stavrakou et al., 2011; Hu et al., 2011; Wells et al., 2014) along with a dramatic underestimate of its free troposphere abundance (Chen et al., 2019) that implies a larger photochemical source than conventionally understood (Muller et al., 2016; Jacob et al., 2005; Bates et al., 2021). The air-sea exchange of methanol also represents a major and long-standing uncertainty in its global budget (Bates et al., 2021; Beale et al., 2013; Yang et al., 2013; Zhou et al., 2023b).

Ethene is a high-yield source of ozone in urban air (Chameides et al., 1992; Ryerson et al., 2003; Zhao et al., 2020), and can contribute to secondary organic aerosol (SOA) formation via its production of glycoaldehyde (Huang et al., 2011). It is also a ubiquitous plant hormone that regulates growth and responds to biotic and abiotic stress (Rhew et al., 2017). The only published global budget assessment for atmospheric ethene concludes that it mainly originates from the terrestrial biosphere (Sawada and Totsuka, 1986). These emissions are highly uncertain as they are generally extrapolated from a single field study over a midlatitude deciduous forest (Goldstein et al., 1996; Guenther et al., 2012); more recent observations imply significantly larger fluxes from coniferous ecosystems (Rhew et al., 2017). Ethene is also widely used in chemical feedstocks, and has the highest production rate of all industrially-generated organic compounds (Rhew et al., 2017). Other sources include fossil fuel and



biomass combustion (Lewis et al., 2013; Ryerson et al., 2003) and agricultural emissions from its use as a ripening agent
(Rhew et al., 2017).

Ethyne and HCN are both emitted during combustion: HCN originates primarily from biomass burning (Li et al., 2003),
whereas ethyne also has an important fossil fuel contribution (Xiao et al., 2007). Since HCN emissions depend on the nitrogen
content of the biomass being burned (Coggon et al., 2016), downwind observations can yield insights into fuel composition.
Once in the atmosphere, HCN plays a role in the nitrogen cycle (Li et al., 2000) while ethyne is a potentially important SOA
source (Volkamer et al., 2009). The distinct sources and relatively long atmospheric lifetimes for these compounds (~5 months
for HCN versus ~2 weeks for ethyne; (Li et al., 2003; Xiao et al., 2007)) make them a useful tracer pair for partitioning biomass
versus urban combustion sources (Crounse et al., 2009) and associated transport pathways (Duflot et al., 2015). Ethyne and
CO observations can further be combined to constrain the photochemical age of air (Swanson et al., 2003; Xiao et al., 2007).

Satellite-based infrared (IR) measurements can yield powerful information on the aforementioned atmospheric VOCs, and
there have been a number of advances on this front in recent years. Space-borne measurements of atmospheric methanol,
ethyne, ethene, and HCN were performed by solar occultation with the Atmospheric Chemistry Experiment - Fourier
Transform Spectrometer (ACE-FTS; Dufour et al., 2006; Coheur et al., 2007; Herbin et al., 2009; Rinsland et al., 2005); HCN
and ethyne were subsequently also retrieved using the Michelson Interferometer for Passive Atmospheric Sounding (MIPAS;
Glatthor et al., 2009; Glatthor et al., 2013; Glatthor et al., 2015; Parker et al., 2011; Wiegele et al., 2012). However, the limb-
viewing geometry of these instruments samples only the upper troposphere and lower stratosphere. Beer et al. (2008) pioneered
the first satellite-based methanol measurements with sensitivity to the lower troposphere using the nadir-viewing Tropospheric
Emission Spectrometer (TES) onboard EOS-Aura. Later work with TES led to the detection and quantification of ethene in
biomass burning plumes (Alvarado et al., 2011; Dolan et al., 2014) and enabled global space-based retrievals of methanol
(Cady-Pereira et al., 2012; Shephard et al., 2015; Wells et al., 2012). However, the coarse TES sampling strategy limited the
spatial information provided.

The Infrared Atmospheric Sounding Interferometer (IASI) onboard the MetOp platforms provides dense daily sampling and
has been used to quantify several atmospheric VOCs. Initial efforts demonstrated detection of methanol, ethene, ethyne, and
HCN (along with other pyrogenic VOCs) in biomass burning plumes (Clarisse et al., 2011; Coheur et al., 2009). Subsequent
work by Razavi et al. (2011) derived global methanol distributions from the IASI radiances using a brightness temperature
difference approach. Global distributions of ethyne and HCN have also been derived from IASI radiances, with sensitivity
mainly in the upper troposphere (Duflot et al., 2013; Duflot et al., 2015). More recent work with IASI has utilized the
hyperspectral range index (HRI) within a machine-learning framework to derive global methanol distributions with greater
sensitivity than previously achieved (Franco et al., 2018). The same methodology has been used with IASI to detect persistent
industrial ethene enhancements (Franco et al., 2022), but no global retrieval of this compound has been published to date.



In this work we present new space-based retrievals for the above VOCs using observations from the Cross-track Infrared Sounder (CrIS) onboard the Suomi-NPP (SNPP) satellite. CrIS has important advantages over other sensors that can enhance our ability to quantify VOCs from space in the thermal IR. First, CrIS provides dense spatial and temporal sampling that is comparable to IASI, but with much lower instrument noise (Zavyalov et al., 2013). Second, its daytime overpass occurs in the early afternoon when surface-atmosphere thermal contrast and vertical mixing is maximized—yielding greater sensitivity to

the lower troposphere and coinciding with the diurnal emission peak for many VOCs. These advantages enabled the development of the first space-based measurements of isoprene from CrIS—initially over the Amazon by optimal estimation (Fu et al., 2019) and then globally through a machine-learning brightness-temperature difference approach (Wells et al., 2020). We have since developed a more sensitive and generalized retrieval framework based on the HRI for fast, daily detection of VOCs: the Retrieval of Organics from CrIS Radiances (ROCR) (Wells et al., 2022). In this work, we extend the ROCR

framework to include methanol, ethene, ethyne, and HCN. We first confirm the spectral detection of these target VOCs in the CrIS radiances, and subsequently apply an updated version of the ROCR algorithm (ROCRv2) to derive global distributions of methanol, ethene, ethyne, and HCN column densities across the CrIS-SNPP record (2012-2023). We evaluate these results against ground-based column observations, and examine the spatial and temporal variability of the retrieved products in terms of the information CrIS provides on VOC sources across the globe.

**2 Retrieval methodology**

   CrIS is a Fourier transform spectrometer currently flying onboard three sun-synchronous polar-orbiting spacecraft: Suomi-NPP (launched 10/2011), NOAA-20 (10/2017), and NOAA-21 (11/2022). CrIS is a scanning instrument with a 14 km nadir footprint and a 2200 km cross-track scan width, enabling near-global coverage twice daily with a ~1330 LT equator overpass in the ascending node. Radiances in the CrIS longwave IR band (650-1095 cm$^{-1}$) have 0.625 cm$^{-1}$ spectral resolution and a low

noise level of ~0.04 K at 900 cm$^{-1}$ and 280 K (Zavyalov et al., 2013). Analyses here focus on Suomi-NPP observations, with future work planned to apply the same retrieval methodology to all three CrIS sensors.

  **2.1 CrIS HRI calculation**

   ROCRv2 analysis begins with single-footprint Level 1B CrIS spectra that we screen for clouds based on the difference between the surface temperature and the CrIS-measured 900 cm$^{-1}$ brightness temperature, following our approach for isoprene (Wells

et al., 2020). We then compute the hyperspectral range index, HRI, for each spectrum and target species. The HRI (Walker et al., 2011) is a dimensionless measure of the spectral signature for a given species. It is derived by weighting the CrIS-observed radiances by the spectral sensitivity to the target species and inversely by the correlations between spectral channels in background conditions (the latter characterizes the effects of temperature, water vapor, surface emissivity, and other interferents), as shown in Eq. 1:



$$\text{HRI} = \frac{1}{N} \frac{K^T S_{\bar{y}}^{-1}(y - \bar{y})}{\sqrt{K^T S_{\bar{y}}^{-1} K}},\qquad(1)$$

Here, $y$ is a measured spectrum and $\bar{y}$ and $S_y$ are respectively the mean background spectrum and covariance matrix, both derived from scenes where the target species is present only at background levels. We calculate these terms on a monthly basis from global CrIS spectra using an iterative approach that first computes the HRI for all spectra in each of 10 view angle bins (Wells et al., 2022) and then successively removes those exceeding a specified HRI threshold (>1.5 here). Under background

conditions, the HRI as defined has a mean of 0.0 and a standard deviation of 1.0. To preserve this behavior while iterating, we calculate a normalization factor N as the standard deviation over the remote Pacific Ocean (10 – 30 ºS, 180 – 130 ºW) where the target species considered here should typically only be present in background amounts.

$K$ is the spectral Jacobian for a change in the target species, which we calculate for each view angle bin using the Line-By-Line Radiative Transfer Model (LBLRTM; Alvarado et al., 2013) with input VOC profiles from the GEOS-Chem chemical

transport model (v13.3.2; www.geos-chem.org). HCN is not a standard GEOS-Chem tracer and was therefore added here following a previously-developed offline simulation (Li et al., 2003; Li et al., 2009). The computed Jacobians reflect mean model output over the southeast US during summer. Sensitivity analyses confirmed that different vertical lofting conditions alter the Jacobian magnitude but not its shape, which is the relevant factor in Eq. 1. Figure 1 shows the resulting normalized Jacobians for HCN, ethyne, ethene, and methanol. For retrievals that follow, we employ narrow spectral ranges encompassing

the peak absorption features for each species: the $\nu_2$ feature (H-C-N bend) at 712 cm$^{-1}$ for HCN, the $\nu_5$ feature (C-H bend) at 730 cm$^{-1}$ for ethyne, the $\nu_7$ feature (=CH$_2$ wag) for ethene at 949 cm$^{-1}$, and the $\nu_8$ feature (C-O stretch) at 1033 cm$^{-1}$ for methanol. We also test the use of wider spectral windows given the possibility that these may better characterize the influence of interferences; both sets of windows are defined in Table 1.

**2.2 Detecting HCN, ethyne, ethene, and methanol spectral signatures in CrIS radiances**

We illustrate the spectral detection of HCN, ethyne, ethene, and methanol in the CrIS radiances by analyzing a plume over the remote Pacific that was emitted and transported from the strong 2019-2020 Australian wildfires. Previous work with IASI has likewise identified enhancements of the target VOCs in these fire plumes (Pope et al., 2021; De Longueville et al., 2021). Figure 2a-d shows that strongly elevated CrIS HRI signals are obtained for all four species in the fire plume observed on 2 January 2020.

To visualize the underlying spectroscopic signal in each case we calculate full-filter CrIS spectral residuals (see Supplementary Information S1 for details). We fit for cloud properties, surface temperature, atmospheric temperature, H$_2$O, CO$_2$, O$_3$, HDO, N$_2$O, CH$_4$, NH$_3$ and PAN. SO$_2$, HNO$_3$, OCS, SF$_6$, HCOOH, CFC-11, CFC-12, and CCl$_4$ are accounted for via climatological profiles. Observed-calculated residuals after the fit reveal the presence of additional absorbers. Figure 2e shows the difference between the mean in-plume and mean out-of-plume residuals computed in this way, with the plume extent defined using the



TROPESS CrIS CO product ($> 5 \times 10^{18}$ molec/cm$^2$; Bowman, 2021). Results show that the four strongest unaccounted-for peaks in the CrIS longwave IR band are associated with our target species, confirming their importance in fire plumes and the underlying spectral signals driving the HRI values.

## 2.3 ANN-based retrieval

We use an artificial neural network (ANN) retrieval approach to convert the CrIS-observed HRI values to VOC column
abundances. The network is trained on a synthetic HRI dataset derived from LBLRTM-generated radiances. These radiances are computed using input VOC profiles from a global GEOS-Chem simulation (see Supplementary Information S2 for details) performed at $2° \times 2.5°$ for 2019, sampled every 16 days with random perturbations ($1\sigma = 100\%$) applied to reduce correlations with co-varying atmospheric variables. Concentrations of other trace gases are from either GEOS-Chem (ammonia, ozone) or a MOZART (Brasseur et al., 1998) climatology (methane, CO$_2$, CFCs); temperature and water vapor are from MERRA-2
reanalysis (Gelaro et al., 2017). Each scene is simulated with and without the target species at three random viewing angles selected from [0°, 16°], [17°, 32°], and [33°, 48°]. LBLRTM spectral output for simulations with the target species are replicated 25 times per scene and view angle, adding random CrIS-like noise each time. The training-set HRI$_{synth}$ values are then calculated as:

$$\text{HRI}_{\text{synth}} = \frac{K^T S_y^{-1}(y - y_o)}{\sqrt{K^T S_y^{-1} K}}, \tag{2}$$

where $y$ and $y_o$ are the LBLRTM radiance spectra with and without the target species, respectively, and $S_y$ is the CrIS-derived background covariance matrix for that month. The final HRI$_{synth}$ value for each scene and view angle is taken as the mean across all 25 replications.

The resulting training set contains 646,272 HRI$_{synth}$-column pairs per species, and these are plotted in Fig. 3. The sensitivity of thermal IR observations hinges on the thermal difference between the Earth's surface and the absorber, and therefore on the
vertical location of the absorber within the atmospheric column. For the purpose of ANN training and predictions we characterize this sensitivity using a predictor hereafter referred to as P$_{90}$: the atmospheric pressure below which 90% of the VOC column resides. We see in Fig. 3 that P$_{90}$ strongly modulates the HRI$_{synth}$-column relationship: detection sensitivity increases as P$_{90}$ decreases (i.e. as the VOC is lofted higher in the atmosphere). This dependence is steepest for HCN and ethyne, which have features in areas with strong CO$_2$ absorption; their sensitivity is therefore strongly enhanced by vertical
displacement relative to that interference. As will be seen, explicit inclusion of P$_{90}$ as an ANN predictor succinctly captures this vertical dependence, and also allows consistent vertical profile assumptions to be applied when comparing the CrIS data to model output or to independent observations.

We next train a feedforward neural network to derive VOC column abundances from the corresponding HRI values while taking into account the relevant factors that affect that relationship. Along with P$_{90}$ we include six predictors describing the



thermodynamic state of the atmosphere (water vapor column, surface skin and surface air temperatures, and atmospheric temperatures at ~850, 625, and 225 hPa), surface pressure, and satellite view angle. Network architecture includes two hidden layers (20 and 10 nodes) employing hyperbolic sigmoid transfer functions and a single-node linear output layer. As in our past work, the training is done on 10 random extractions of the data, with the final prediction taken as the mean of those 10 networks.

Figure 4 summarizes the resulting ANN performance, precision (calculated as the relative standard deviation, RSD, across the

10 networks), and bias (relative to the true input values). The 10 network-mean reproduces 96% of the training set column variance for HCN, 94% for ethyne and ethene, and 97% for methanol. In all cases the precision is high (RSD < 15%) for elevated column abundances; lower columns (< $2\times10^{15}$ molec/cm$^2$) exhibit degraded precision (RSD > 60%) with varying dependence on thermal contrast ($\Delta T$). Precision for HCN and ethyne shows little $\Delta T$ dependence, with the RSD increasing only slightly to ~20% for $\Delta T > 15$ K (and a corresponding 20% high bias in these conditions). Ethyne predictions also exhibit

a slight low bias (5-10%) for columns > $2\times10^{15}$ molec/cm$^2$ that manifests at all $\Delta T$ values. Prediction performance has the strongest $\Delta T$ dependence for ethene: here, when $\Delta T < 0$ the RSD ranges from 20-40% with a significant (30-50%) low bias— compared to ~20% RSD (with a 10% high bias) when $\Delta T > 15$ K. Methanol generally exhibits the highest prediction precision (RSD < 5%); the RSD increases slightly to 10% for $\Delta T > 10$ K and to 20-30% for $\Delta T < 0$ and columns < $1\times10^{16}$ molec/cm$^2$ (there is also a 30% high bias under these last conditions).

Variable withholding tests show that for all four species the HRI is the most important input variable for the resulting column predictions, and this is especially the case at higher column amounts (Fig. S1). $P_{90}$ is next in importance, confirming that the measured signal is dominated first by the species abundance and second by its vertical location in the atmosphere. Withholding the $P_{90}$ predictor yields larger error increases for HCN and ethyne than for methanol and ethene, reflecting the stronger vertical sensitivity and $CO_2$ impacts on the former two species. The other predictors each wield a smaller influence on the retrieval,

with their relative importance varying somewhat by species. The influence of atmospheric temperature, in particular, is dispersed among multiple input variables that individually have only minor impacts on the retrieval.

Finally, the 10 ANNs are applied to the CrIS HRIs in combination with assimilated meteorological observations (water vapor, temperature) from MERRA-2. The resulting output represents the VOC column enhancements above their climatological background; as discussed later these backgrounds must then be incorporated to obtain the absolute column amounts. We

perform the retrieval for each VOC across a fixed set of $P_{90}$ values applied globally. Those $P_{90}$ values were chosen to span the statistical distribution obtained from global GEOS-Chem predictions for each species, and span 150-650 hPa for ethyne and methanol, 50-650 hPa for HCN, and 250-850 hPa for ethene; interpolation can then be performed between the two proximal values when and where observational constraints are available.

The spread in results obtained by applying the 10 individual ANNs to the CrIS radiances provides an additional measure of

confidence in the satellite-derived columns, and can be used to help flag retrievals that are less robust due to low measurement sensitivity and/or greater influence from interferences. Figure 5 plots this observed spread as the RSD across the 10-ANN




ensemble for each CrIS retrieval, binned as a function of VOC column amount and thermal contrast. A uniform vertical profile assumption is employed here for each VOC (global median $P_{90}$ = 150 hPa for HCN, 350 hPa for ethyne and methanol, and 550 hPa for ethene). For ethyne, ethene, and methanol, the RSD is large under negative thermal contrast conditions, as expected

based on the retrieval performance discussion above and thermal infrared sensitivity in general. There is also substantial ANN spread for HCN and ethyne for certain low thermal contrast, high abundance (> $4\times10^{16}$ molecules/cm$^2$) retrievals, and at very high thermal contrast for ethene: these represent exceedingly rare events that are only sparsely represented in the training set. Outside of these conditions, the 10-ANN RSD is generally <20%, and we use this threshold as a guideline for screening the results moving forward. We retain cases that naturally have high RSD due to low column amounts by also requiring the 10-

ANN absolute SD to exceed its 75$^{th}$ percentile for any data removal to occur. Lastly, we screen scenes with surface temperatures < 270 K, which are low sensitivity conditions not represented in our training set. Figure S2 shows that most such scenes exhibit large observational uncertainties, particularly for ethene and methanol.

## 3 VOC distributions seen from CrIS

### 3.1 Global and seasonal variability

Figure 6 maps the seasonal mean HCN, ethyne, ethene, and methanol columns (as above-background enhancements) derived using the ROCRv2 retrieval and averaged across the 2012-2023 SNPP-CrIS record. Monthly-mean timeseries are also shown in Figures 7 and 8 for select regions defined in Fig. S3. Each of these plots employs a globally-fixed vertical profile assumption based on the GEOS-Chem-predicted global median $P_{90}$ for each species: 150 hPa for HCN, 350 hPa for ethyne and methanol, and 550 hPa for ethene. While the real $P_{90}$ values will vary in space and time with the meteorological and source factors that

influence vertical tracer distributions, the fixed-$P_{90}$ columns are useful for identifying the patterns of variability that are actually detected by the satellite measurement (and not imposed by differing model-assumed profile shapes).

The maps in Figure 6 reveal robust global distributions that reflect emissions and transport from known biogenic, biomass burning, and anthropogenic source regions. The HCN and ethyne distributions exhibit many similarities to those presented for IASI in Duflot et al (2015), while the CrIS methanol fields are broadly consistent with published IASI results for 2011 (Franco

et al., 2018). The ethene maps in Figure 6 represent the first global column distribution derived from space for this species; the principal hotspots are consistent with those featured in the regional analyses by Franco et al. (2022). Below, we highlight some of the key features revealed in the CrIS column distributions.

### 3.1.1 December, January, February (DJF)

Strong HCN, ethyne, methanol, and (to a lesser extent) ethene column enhancements are present each year during the DJF

biomass burning season over equatorial Africa (Figure 6 and 7a). The methanol enhancements extend further north into the Sahel, where previous work found biogenic methanol emissions to be underestimated (Wells et al., 2014). Methanol enhancements are also detected over the southeastern Amazon basin and southern Africa (Angola/Zambia); these are most





likely biogenic in origin given their southerly displacement from the biomass burning tracers. The latter region was likewise revealed by CrIS to be an isoprene hotspot at this time of year (Wells et al., 2020). A weaker methanol hotspot is seen during
this season over northern Australia (Fig 8c), where Stavrakou et al. (2011) identified significant biogenic emissions.

Large ethene enhancements are evident over eastern China in the vicinity of Beijing; this wintertime signal is an annual feature in the CrIS record (Fig. 8a) and occurs in the same area where the highest ethene HRIs were detected by IASI and attributed to coal and petrochemical emissions (Franco et al., 2022). Wintertime HCN enhancements are observed over this same region by CrIS but may reflect industrially-emitted interferences as discussed in the next section. Finally, high-latitude enhancements
of HCN and ethene are visible during DJF over the North Atlantic and Southern Oceans; we interpret these as potential artifacts arising from surface effects due to sea ice or whitecaps in high winds, but more work is needed to confirm this.

### 3.1.2 March, April, May (MAM)

The biomass burning signals seen over tropical Africa during DJF persist into MAM, when strong HCN, ethyne, ethene, and methanol signals are also detected over southeast Asia. Transpacific transport from the latter region is evident in Figure 6 for
the longer-lived species (HCN, ethyne, and methanol). These transported fire emissions are detected each spring at Mauna Loa (see Section 4) and may contribute to the springtime HCN and ethyne peaks observed by CrIS each year over the continental US (Fig. 7e). HCN, ethene, and methanol enhancements over southern Russia and northern China coincide with the regional springtime peak in fire activity (Giglio et al., 2006), though early-season biogenic emissions likely also contribute for ethene and methanol. Methanol columns over equatorial and Northern Africa peak during this season every year (Fig. 7a), when
temperatures and isoprene emissions are also highest (Marais et al., 2014). The ROCRv2 retrievals also reveal methanol and ethene enhancements over Brazil during this season that are likely biogenic given the lack of co-occurring peaks in HCN and ethyne.

### 3.1.3 June, July, August (JJA)

The HCN, ethyne, ethene, and methanol hotspots seen by CrIS over Africa shift south at this time of year, peaking over the
Democratic Republic of Congo. Significant enhancements over India and southeast Asia are seen for ethyne and, to a lesser extent, for HCN and methanol. These signals reflect biomass burning and industrial activities and, given the use of globally-fixed $P_{90}$ values in Figures 6-8, are presumably enhanced by efficient vertical lofting during the summertime South Asian Monsoon (e.g., Lelieveld et al., 2018). HCN, ethyne, and ethene enhancements detected by CrIS over Canada and Russia during JJA largely correspond to wildfire activity, with widespread methanol enhancements over southeastern Russia/northern
China and eastern US/Canada also reflecting biogenic emissions. The Russia/northern China region exhibits a summer methanol peak every June across the CrIS record (Fig. 8d), whereas dual spring-summer peaks are often detected over the SE US (Fig. 7e)—perhaps reflecting competing effects from increasing leaf area index over the growing season versus higher methanol emission factors from younger leaves (Wells et al., 2012). Over Amazonia, the ROCRv2 methanol retrievals reveal a seasonal minimum in JJA with the widespread leaf flushing that occurs between the regional wet and dry seasons (Barkley



et al., 2009). Elevated VOC columns retrieved over the far Southern Ocean are likely due to wintertime surface artifacts, as previously discussed.

### 3.1.4 October, November, December (OND)

Major enhancements are detected for all four VOCs over Amazonia, southern Africa, and Indonesia during the SON biomass burning peak that manifests in these regions. The ROCRv2 VOC timelines over Indonesia reveal particularly dramatic peaks during the unprecedented 2015-2016 fire season that was fueled by an El Niño-induced drought (Fig. 7b; Field et al., 2016). The measured HCN columns over India and Australia also peak in 2016 due to transport from those fires. Efficient long-range transport from South America to southern Africa, and from southern Africa to Australia, is evident in Fig. 6 for the longer-lived species (HCN, ethyne, methanol) at this time of year. These transport pathways have been identified in other satellite-based studies (Gloudemans et al., 2006; Rinsland et al., 2005), and drive much of the seasonality in fire-related air pollution over New Zealand (Edwards et al., 2006) and Australia (Fig. 8c).

### 3.2 Vertical sensitivity of measurement

While the results above employ a single vertical profile shape ($P_{90}$ value) for each species, the ROCRv2 retrieval is performed for a range of $P_{90}$ values to explicitly represent the observational sensitivity to this factor and to enable customization via interpolation. Figure 9 shows how the retrieved column varies with assumed $P_{90}$ over Amazonia (region defined in Fig. S3) during the dry season when signals are highest due to widespread burning and strong biogenic emissions. The strength of the column-$P_{90}$ relationship reflects the underlying spectroscopy for each species, with HCN and ethyne exhibiting stronger vertical sensitivity due to overlapping $CO_2$ absorption. For example, the computed HCN and methanol columns both vary by a factor of 1.6 across the relevant $P_{90}$ range—despite the fact that this range spans ~250 hPa in the case of methanol versus just ~100 hPa for HCN. Ethyne varies by over 5× across the regional $P_{90}$ range simulated by GEOS-Chem for this species (~300 hPa), whereas ethene varies by only 2× across a $P_{90}$ range of ~450 hPa.

Ideally one would use an observationally constrained $P_{90}$ value to interpret the CrIS output at a given place and time. However, this is not usually available and one may instead choose to use a model estimate of this quantity, which then also enables an internally consistent satellite-model comparison. Figure 10 provides an illustration of this vertical adjustment in terms of its effects on the derived VOC spatial distributions. Data are plotted for July 2019 and either i) employ a globally-fixed $P_{90}$ value, or ii) employ GEOS-Chem to estimate the local $P_{90}$ value (mapped in Fig. S4) for each individual CrIS retrieval. Accounting for spatial variability in the VOC spatial distributions generally reduces the derived columns over background and long-range transport regions while increasing them over VOC source regions. Results over biogenic source regions show that this adjustment is typically smaller for methanol than it is for ethene—due to lower vertical profile sensitivity for methanol, plus (as will be seen) an unrealistic modeled profile shape for ethene in such regions.



When the VOC profile shape is poorly known, retrieval uncertainty will be correspondingly higher; the effect will be largest
for HCN and smallest for methanol based on the vertical sensitivities discussed earlier. This issue manifests in particular for
fires due to variability and uncertainty in the vertical lofting of emissions. When employing an alternative biomass burning
inventory (GFAS; Di Giuseppe et al., 2018) in GEOS-Chem for $P_{90}$ computation, we indeed see some positive and negative
differences in the CrIS-retrieved columns over fire regions relative to the base-case using GFED4 (van der Werf et al., 2017;
Fig. S5). These changes arise from differing plume injection assumptions between these two inventories (Jin et al., 2023) but
also from the divergent fire emission magnitudes which themselves modify the VOC vertical profiles and thus the derived $P_{90}$
values. These sensitivities emphasize the need for consistent profile assumptions when interpreting and comparing the CrIS
data, and the importance of diagnosing the associated uncertainty space.

## 4 Retrieval evaluation with NDACC data

We evaluate the CrIS column retrievals using ground-based Fourier transform infrared (FTIR) solar absorption measurements
from sites in the Network for the Detection of Atmospheric Composition Change (NDACC; De Mazière et al., 2018). NDACC
FTIR instruments measure direct solar absorption spectra under clear-sky conditions; trace gas profiles are then retrieved via
optimal estimation (Rodgers, 2000). Figure S6 maps the sites used in this work, which include 14 established NDACC stations
and two candidate stations (Xianghe, China; Zhou et al., 2023a and Porto Velho, Brazil; Vigouroux et al., 2018). Spectral
microwindows used in the NDACC retrievals of HCN, ethyne, and methanol are listed in Table 2. We also analyzed ethene
measurements at Jungfraujoch, and Maido; however, we omit them here due to low column abundances ($< 6 \times 10^{14}$ molec/cm$^2$)
at these sites plus representation uncertainty arising from the elevation difference between the observatories and the
surrounding CrIS footprint. The NDACC methanol retrievals employ spectral features that overlap with those used for CrIS
whereas the HCN and ethyne retrievals use mid-wave IR features. Reported random and systematic errors for the NDACC
VOC retrievals are generally <15% (Table 2), with one study reporting a higher random error for ethyne (~34%; Yamanouchi
et al., 2023). More details on the NDACC retrievals can be found in Vigouroux et al. (2012), Bader et al. (2014), Ortega et al.
(2021), Zhou et al. (2023a), and Yamanouchi et al. (2023).

Figure 11 shows CrIS vs. NDACC comparisons for the ensemble of sites measuring methanol, ethyne, and HCN. The CrIS
retrievals are screened as described in Section 2.3 and interpolated to the local $P_{90}$ value as computed from the NDACC-
retrieved VOC profile, thus ensuring a consistent vertical distribution assumption between the two datasets. In cases where the
NDACC site lies at a significantly higher altitude than that of the 0.5° × 0.625° CrIS pixel, we scale the CrIS value according
to the NDACC/CrIS overhead column fraction predicted by GEOS-Chem for each location and time. Those scaling factors are
plotted in Fig. S7; for HCN and methanol they vary seasonally, with minima during summer when pyrogenic and biogenic
emissions result in steeper vertical gradients.



The CrIS and NDACC methanol retrievals are strongly correlated on both daily (r=0.77) and monthly (r=0.84) timescales (Fig. 11a and b), showing that atmospheric methanol variability is well-captured by the ROCRv2 retrieval. The offset between the two datasets is expected since the CrIS data represent above-background enhancements; using the x-intercept of the CrIS/NDACC fit to estimate this background yields a value of $1.1 \times 10^{16}$ molec/cm$^2$ (daily comparison; 95% CI: $1.0 - 1.2 \times 10^{16}$ molec/cm$^2$) or $7.1 \times 10^{15}$ molec/cm$^2$ (monthly means; 95% CI: $6.2 - 7.9 \times 10^{16}$ molec/cm$^2$). After adding these background

values to the CrIS column enhancements, we obtain a normalized mean bias (NMB) for the CrIS methanol retrieval of -7.0% based on the daily comparison or -17% based on the monthly comparison.

Residual disparities may reflect some systematic uncertainty in the comparison. Timelines for individual sites with long-term methanol records (Fig. S8) reveal that the apparent 7-17% low bias for CrIS relative to NDACC is driven most strongly by the St. Petersburg comparison, where CrIS exhibits a 32% low bias. At other sites, the background-corrected normalized mean

biases for CrIS with respect to NDACC fall within the NDACC-reported systematic uncertainty range. The strong CrIS vs. NDACC correlation seen above in the ensemble comparison also manifests at the individual sites, with the exception of Jungfraujoch (r=0.31). The poor Jungfraujoch correlation partly reflects uncertainty in the GEOS-Chem derived altitude correction for this high-elevation site, which is compromised by the model's known methanol underestimate in the free troposphere (Chen et al., 2019; Bates et al., 2021). When the altitude correction is omitted we instead obtain r=0.56 at this site

(monthly means).

The CrIS/NDACC ethyne comparison (Fig. 11c and d) shows a moderate correlation (r=0.56 and 0.65 for daily and monthly means) with an offset based on reduced major axis regression of $3.8 \times 10^{14}$ molec/cm$^2$ [95% CI: $2.9 - 4.7 \times 10^{14}$ molec/cm$^2$] for the daily comparison and effectively zero [95% CI: $-1.5 - 2.7 \times 10^{14}$ molec/cm$^2$] for the monthly comparison. If we define an ethyne background based on the daily CrIS/NDACC comparison intercept we obtain a normalized mean bias (NMB) for

the resulting CrIS ethyne fields of -40% with respect to NDACC. For HCN, the correlations with NDACC are lower (r=0.36-0.44; Fig. 11e and f); however, adding the corresponding background offsets (daily: $3.5 \times 10^{15}$ molec/cm$^2$ [95% CI: $3.4 - 3.5 \times 10^{15}$ molec/cm$^2$]; monthly: $3.2 \times 10^{15}$ molec/cm$^2$ [95% CI: $3.2 - 3.3 \times 10^{15}$ molec/cm$^2$]) to the CrIS data implies an NMB of just -2 to +5%, well within the NDACC-reported uncertainties.

Site-specific timelines for ethyne (Fig S9) reveal a CrIS/NDACC correlation of r = 0.57 at Xianghe, the site with highest

column concentrations. At Toronto the correlation is lower (r =0.43), with CrIS capturing the seasonal peak detected by NDACC in some years but not others. HCN also exhibits site-dependent performance: CrIS observations in the tropics and Southern Hemisphere (Fig. S10) show higher correlations with NDACC (r=0.44-0.70) than is obtained in the Northern Hemisphere midlatitudes, where correlations are sometimes negative—particularly at Xianghe (Fig. S11). However, in the tropics, the CrIS HCN observations capture the NDACC-detected seasonal biomass burning peaks well (aside from one event

at the end of 2015).





The weaker CrIS/NDACC agreement for HCN and ethyne is likely due at least in part to their strong sensitivity to the assumed vertical profile shape, combined with the fact that the NDACC retrievals do not adequately constrain the true $P_{90}$ values. At Toronto, for example, NDACC mean degrees of freedom for signal (DOFS) are 1.5 for methanol, 1.4 for ethyne and 2.1 for HCN (Yamanouchi et al., 2023), indicating that the retrievals provide no more than two pieces of vertical information and that

the retrieved $P_{90}$ is therefore heavily informed by the prior. Figure S12 shows that repeating the CrIS/NDACC comparisons using GEOS-Chem-derived rather than NDACC-derived $P_{90}$ values has little impact on the conclusions, although the modeled vertical profile shapes have their own uncertainties.

Some of the above results are sensitive to the choice of spectral window used in the CrIS retrievals. In particular, the negative CrIS/NDACC correlation seen at Xianghe for HCN is reversed when a wider spectral range is used in the HRI derivation (Fig.

S13b). In both cases the resulting CrIS column variability closely tracks that of the underlying HRI, showing that the opposing seasonality at this site arises from a spectral interference rather than being imposed by one of the other ANN predictors. Sensitivity tests rule out $NO_2$ interference as the cause; further work is needed to determine whether particular trace gases or surface interferences are affecting the HCN retrieval at Xianghe and potentially other urban areas.

## 5 Global VOC distributions from CrIS compared to model predictions

The ROCRv2 VOC datasets provide new and complementary constraints for characterizing biomass burning, biogenic, and industrial sources, tracking the evolution and fate of emitted compounds, and testing model representations of these processes. Figures 12 and 13 compare the CrIS VOC measurements to predictions from the GEOS-Chem CTM (configured as described in Supplementary Information) for January, April, July, and October 2019. The CrIS retrievals are adjusted to the simulated $P_{90}$ values for each species and have a constant increment added to represent the global background as determined from the

NDACC comparisons above ($3.5 \times 10^{15}$ molec/cm$^2$ for HCN, $3 \times 10^{14}$ molec/cm$^2$ for ethyne, $1.1 \times 10^{16}$ molec/cm$^2$ for methanol). Given the short lifetime for ethene (~1 day at $10^6$ molec OH/cm$^3$; (Atkinson and Arey, 2003) its expected background levels are below the CrIS limit of detection and no increment is added in this case. Airborne measurements from the ATom campaigns support this treatment, with a mean observed column in the remote atmosphere (outside of fire plumes) of ~$4 \times 10^{13}$ molec/cm$^2$. Model output in Figs. 12-13 is sampled at the CrIS overpass time (1200-1500 LT mean) and has been

subjected to the same data screening described earlier for CrIS data (Section 2.3). Below we highlight some of the main similarities and differences between the two datasets in terms of underlying processes and key aspects meriting further science investigation.

### 5.1 Methanol

GEOS-Chem generally underpredicts the methanol columns measured from space by CrIS, while capturing the main spatial

and seasonal patterns of variability that are observed. Discrepancies occur over East Asia and India during winter, where CrIS reveals methanol hotspots that are absent from the simulations and that point to underestimated and/or missing anthropogenic



sources. The methanol column abundance and seasonality over biogenic source regions is also not well represented in the model: observed springtime enhancements over the southeast US—driven by strong emissions from young leaves (Wells et al., 2012)—are underestimated, while Amazonian emissions are overestimated in July during the leaf flushing period between

wet and dry seasons (Barkley et al., 2009). A substantial model underestimate over southern Africa in January and October (and over Australia in April and October) likely also reflects missing biogenic emissions given the lack of co-occurring enhancements for the biomass burning VOCs.

We also see large satellite-model differences over boreal and tropical burning regions that point to underestimated emissions or secondary production in those plumes. Uncertainties in the fire plume injection heights (and thus in the assumed $P_{90}$ values

applied to the CrIS data) may also play a role here. Finally, the model-predicted methanol background is lower than seen in the CrIS data (and as constrained via the NDACC comparisons)—and this is especially true at high latitudes. Bates et al. (2021) likewise found the remote methanol abundance predicted by GEOS-Chem to be biased low with respect to airborne measurements, suggesting a missing chemical source in its global budget.

### 5.2 Ethene

The ethene columns predicted by GEOS-Chem are broadly consistent with those observed by CrIS over industrial areas of eastern China, Europe, and over the eastern US in the wintertime. The seasonal high-latitude enhancements associated with long-range transport from these sources are also captured, as are the observed enhancements over boreal and tropical African fires. However, major satellite-model discrepancies are seen over biogenic source regions in South America, the southeastern US, northern Australia, and Indonesia. These differences point to a substantial high bias in the modeled biogenic ethene

emissions. Those emission estimates are extrapolated from a single field study over a mid-latitude forest (Goldstein et al., 1996; Guenther et al., 2012); the CrIS data indicate a strong need to characterize ethene emissions over other ecosystems. An additional discrepancy manifests over the Southern Ocean, where the CrIS ethene columns are enhanced (particularly in July) but the model values are not. A slight July ethene peak was also detected in the southern high-latitude upper troposphere by ACE-FTS (Herbin et al., 2009), which could reflect the same phenomenon. However, surface interference from winds or ice

may also be affecting the CrIS ethene retrievals in these areas, and more work is needed to identify the source of this signal.

### 5.3 Ethyne

The GEOS-Chem ethyne predictions show marked spatial consistency with the CrIS observations throughout the year, though with some differences in magnitude. The simulated ethyne columns are lower than CrIS over northwestern South America and tropical Africa during April and over Brazil during October, which may indicate a general biomass burning underestimate for

these regions. Modeled enhancements over industrialized areas in East Asia exceed those seen from CrIS during part of the year, which could reflect a misrepresentation of seasonal anthropogenic sources in GEOS-Chem and/or the possible CrIS low bias for ethyne identified earlier (Section 4). The model also predicts a stronger interhemispheric gradient during much of the



year than is evident in the CrIS measurements, which is consistent with an anthropogenic emission overestimate. A similar interhemispheric gradient mismatch was noted previously for ethyne using satellite-based measurements from IASI ethyne (Duflot et al., 2015).

## 5.4 HCN

The HCN spatial distribution predicted by GEOS-Chem is generally coherent with the CrIS observations. However, there are two wintertime enhancements in the CrIS data that the model does not reproduce: (1) over (and downwind from) East Asia, and (2) over high-latitude oceans. The latter is potentially due to surface interferences associated with ice and/or high winds, as mentioned in Section 3. The former may reflect an unidentified anthropogenic interferent, as these wintertime enhancements are not seen in the ground-based FTIR measurements at Xianghe (Section 4). However, one analysis (Li et al., 2003) found that including an HCN source from Asian coal burning (amounting to ~25% of the global budget) yielded much better model agreement with downwind aircraft observations compared to simulations that only included biomass burning HCN emissions. Further work is needed to elucidate the importance of this potential source, which was not included in the GEOS-Chem simulations used here.

Over biomass burning regions, GEOS-Chem tends to predict higher HCN columns than are seen by CrIS, particularly over boreal regions during northern summer and over Indonesia during October. Conversely, as with ethyne, the simulated HCN columns are lower than CrIS observations over South America and Africa during April and over Brazil during October, pointing to a potential HCN emission underestimate for these locations and seasons. Vigouroux et al. (2012) also found evidence of a tropical HCN emission underestimate in GEOS-Chem (with an earlier GFED version) based on comparisons to FTIR measurements. Akagi et al. (2011) note large variability in observed HCN emission factors across fuel types and fire conditions; the low emission factors for tropical forests reported in their work (and used here) may warrant revision.

## 6 Conclusions

In this work we updated the Retrieval of Organics from CrIS radiances (ROCRv2) algorithm to enable fast, global space-based measurements of methanol, ethene, ethyne, and HCN. We first confirmed the spectral detection of our four target species in the CrIS radiances, and then applied the new algorithm to derive methanol, ethene, ethyne, and HCN column abundances across the entire SNPP-CrIS record (2012-2023). The updated machine learning-based approach explicitly accounts for the dependence of the HRI-column relationship on the VOC profile shape via the $P_{90}$: the pressure below which 90% of the column resides. This dependence is particularly strong for HCN and ethyne given their spectral overlap with $CO_2$ absorption features. The resulting ROCRv2 ANN reproduces 94-97% of the training set variance for each VOC and exhibits a high degree of precision. There is a slight (5-10%) negative training bias for ethyne, with more substantial negative biases for ethene and methanol under low-sensitivity conditions ($\Delta T<0$).



We then apply the ROCRv2 retrieval to the CrIS-observed radiances for a set of globally-fixed $P_{90}$ values; interpolation between these values then allows the user to adapt the retrievals to any observational or model-derived $P_{90}$ estimates for
internally consistent comparisons. Relying on model-estimated VOC profile shapes to inform the CrIS retrievals will necessarily introduce biases where those model $P_{90}$ fields are incorrect. However, to a significant extent model $P_{90}$ biases reflect errors in the location and magnitude of emissions, and can thus be mitigated in source quantification studies by iteratively adjusting the CrIS retrieval to the model $P_{90}$ values as emissions are updated (Brewer et al., 2024).

The ROCRv2 methanol columns agree well with ground-based observations from NDACC (r=0.77-0.84), with an NMB of -7
to -17% after accounting for the methanol column background. The HCN and ethyne observation from CrIS exhibit lower correlations with the NDACC columns, likely reflecting greater sensitivity to the assumed vertical profile shape for these species and the weak constraints on that shape provided by the NDACC retrieval. The background-adjusted CrIS HCN columns exhibit negligible bias with respect to NDACC, whereas the CrIS/NDACC ethyne comparison suggests a ~40% disparity between the two datasets. An apparent urban HCN interference (inferred from observations at Xianghe, China) calls for further
investigation.

The global HCN, ethyne, ethene, and methanol distributions detected by CrIS reveal robust features associated with emissions and transport from known biogenic, biomass burning, and anthropogenic source regions. The corresponding 2012-2023 timeseries characterize the seasonal cycles for those sources, along with their interannual variability and the impact of anomalous events such as the 2015-2016 El Niño. The diverse source contributions and atmospheric lifetimes across the four
target VOCs make them highly complementary for process investigation. An initial comparison of the CrIS observations with model predictions from the GEOS-Chem CTM indicates major gaps in our current understanding of biogenic ethene sources. The comparisons also point to uncertainties in the HCN and ethyne emission factors used to model tropical fires, and to a persistent missing methanol source despite recent advancements (Bates et al., 2021).

The 2012-2023 VOC columns presented here represent the full extent of the S-NPP data record, which lost its longwave IR
channels in August 2023. However, future work will apply the ROCRv2 algorithm to the other CrIS instruments on the Joint Polar Satellite Series platforms, which will continue to offer data coverage to 2030 and beyond and will provide a wealth of information on topics including surface-atmospheric exchange, wildfires, urban emissions, and the resulting impacts on atmospheric composition in a changing climate.

**Code availability**

LBLRTM code is available at https://github.com/AER-RC/LBLRTM. GEOS-Chem code is available at www.geos-chem.org.



**Data availability**

The CrIS Level 1B data used in this work are publicly available at https://snpp-sounder.gesdisc.eosdis.nasa.gov/data/SNPP_Sounder_Level1/SNPPCrISL1BNSR.2/. The ROCRv2 VOC retrievals developed in this work are available now at https://z.umn.edu/ROCRv2_VOCs, and will be placed in a permanent public repository with doi prior to publication. The NDACC data are available at https://ndacc.larc.nasa.gov, or by contacting the individual site PIs.

**Author contribution**

Conceptualization: DBM, KCW, VHP; Data curation: KCW, CV, NJ, EM, MM, TN, IO, MP, KS, MS, DS, RS, MZ; Formal analysis: KCW, VHP, JFB, SK, KECP; Funding acquisition: DBM, KCW, VHP, KECP; Investigation: KCW; Methodology: KCW, DBM, VHP, KECP, RP, JFB, SK; Project administration: DBM; Resources: CV, NJ, EM, MM, TN, IO, MP, KS, MS, DS, RS, MZ; Software: RP, JFB, KCW; Supervision: DBM; Writing – original draft preparation: KCW, DBM; Writing – review and editing: All authors.

**Competing interests**

Some authors are members of the editorial board of AMT.

**Acknowledgments**

This research was supported by the National Aeronautics and Space Administration (NASA; Grant #80NSSC21K1975), the National Oceanic and Atmospheric Administration (NOAA; NA22OAR4310200), and the Minnesota Supercomputing Institute (MSI). Part of this work was carried out at the Jet Propulsion Laboratory, California Institute of Technology, under contract to NASA (80NM0018D0004). We sincerely thank Thomas Blumenstock, Jim Hannigan, Frank Hase, Omaira Garcia, Martine De Mazière, Isamu Morino, Clare Murphy, Justus Notholt, and John Robinson for their contributions to the NDACC data used in this work. Funding via Helmholtz ATMO programme has enabled the sustained NDACC FTIR activities at Izaña since the late 1990s. In addition, the Izaña NDACC FTIR observations have been strongly supported (facilities and operational activities) by the Izaña Atmospheric Research Centre of the Spanish Weather Service (AEMET). Saint-Petersburg State University is acknowledged for support of the St. Petersburg FTIR measurements under research project 123042000071-8 (GZ_MDF_2023-2, PURE ID 93882802). The Toronto FTIR measurements were made at the University of Toronto Atmospheric Observatory (TAO), which has received support from CFCAS, ABB Bomem, CFI, CSA, ECCC, NSERC, ORDCF, PREA, and the University of Toronto. The Xianghe FTIR measurements are supported by the National key research and development program (2023YFB3907505).



525

| Species | Narrow spectral range (cm⁻¹) | Wide spectral range (cm⁻¹) |
|---|---|---|
| **HCN** | 700-720 | 690-800 |
| **Ethyne** | 720-740 | 700-800 |
| **Ethene** | 940-960 | 870-1030 |
| **Methanol** | 1020-1040 | 970-1080 |

**Table 1. Narrow and wide spectral ranges considered in this work for the CrIS retrievals.**

| Species | Sites | Microwindows (cm⁻¹) | Systematic error (%) | Random error (%) |
|---|---|---|---|---|
| **Methanol** | Jungfraujoch, Maido, Porto Velho, St. Petersburg, Toronto | 992.00-998.7, 1029.00-1037.00 | 9-15 | 4-10 |
| **Ethyne** | Boulder Jungfraujoch, Maido, Porto Velho, Toronto, Wollongong, Xianghe | 3250.43-3250.77, 3255.18-3255.73, 3268.25-3268.75, 3304.83-3305.35 | 6-11 | 8-34 |
| **HCN** | Boulder, Bremen, Jungfraujoch, Lauder, Maido, Mauna Loa, Paramaribo, Porto Velho, Rikubetsu, St. Denis, St. Petersburg, Izaña, Toronto, Wollongong, Xianghe, Zugspitze | 3268.05-3268.40, 3287.10-3287.35, 3299.40-3299.60, 3331.40-3331.80 | 4-14 | 3-5 |

530 **Table 2. NDACC sites, retrieval microwindows, systematic errors, and random errors as reported in Vigouroux et al. (2012), Bader et al. (2014), Ortega et al. (2021), Zhou et al. (2023a), and Yamanouchi et al. (2023). Not all microwindows were used at all stations.**





535

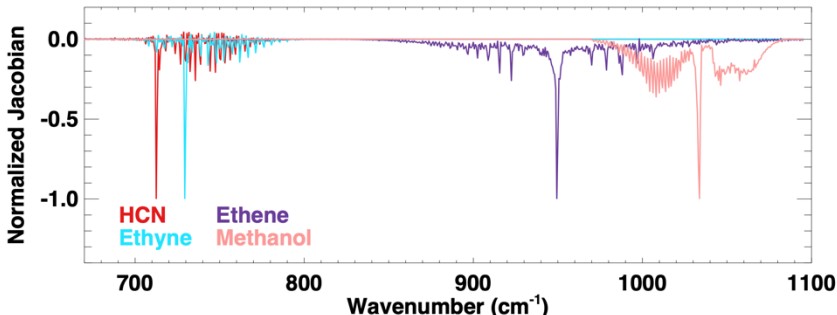

**Figure 1: Normalized spectral Jacobians for HCN, ethyne, ethene, and methanol. Each Jacobian represents the top-of-atmosphere brightness temperature sensitivity to a change in absorber column density. Results are shown for a satellite view angle of 2.5° and were derived using LBLRTM and model-simulated VOC profiles. Each plotted curve has been normalized by its peak value.**



**Figure 2: HCN, ethyne, ethene, and methanol spectral signals as measured from space by CrIS. (a-d) Elevated HRI values for all four species observed on 2 January 2020 in a transported Australian fire plume over the southern Pacific Ocean. (e) Mean in-plume – out-of-plume difference in the TROPESS brightness temperature residuals for the same time and domain, showing the underlying spectral features in each case. See text for details.**



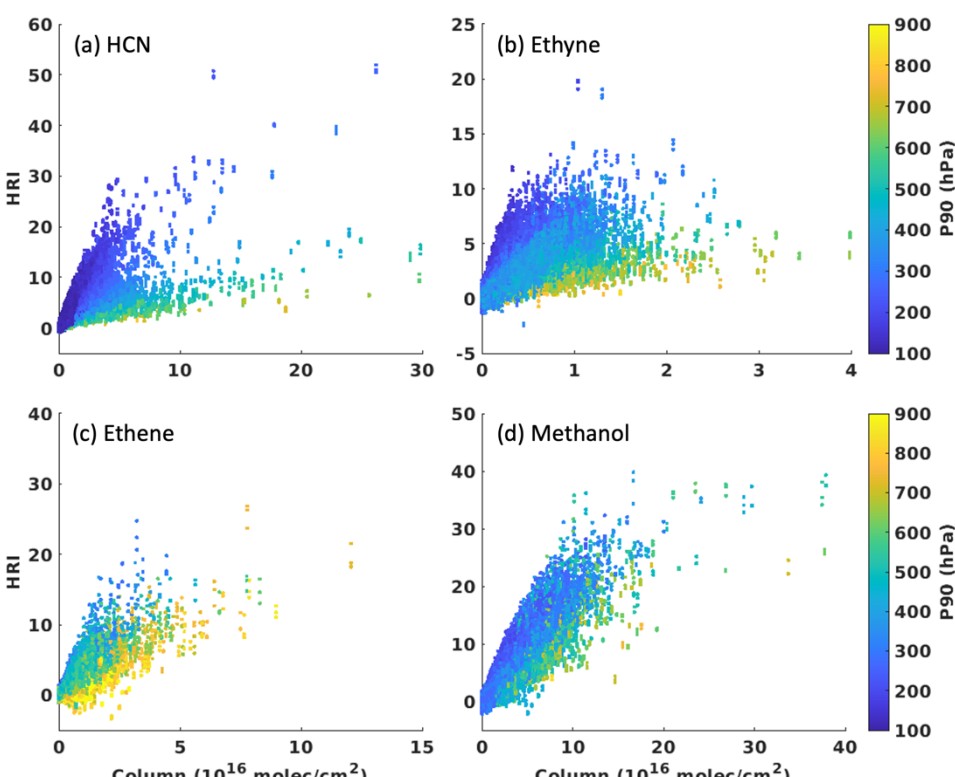

**Figure 3: ANN training sets used in the ROCRv2 VOC retrievals, illustrating the HRI-column relationship and the vertical dependence of detection sensitivity. Data are plotted for (a) HCN, (b) ethyne, (c) ethene, and (d) methanol, and are shaded by $P_{90}$, the atmospheric pressure below which 90% of the VOC column resides.**





**Figure 4: Training performance for the ANNs used in the ROCRv2 VOC retrievals from CrIS.** (a-d) ANN-predicted versus true columns for HCN, ethyne, ethene, and methanol, displayed as the mean (red points) and standard deviation (blue error bars) across the 10 underlying networks. (e-h) Prediction precision (calculated as the relative standard deviation across the 10 networks) and (i-l) bias in the predicted columns with respect to the true input values, binned as a function of thermal contrast (ΔT, surface skin temperature – surface air temperature) and column amount.





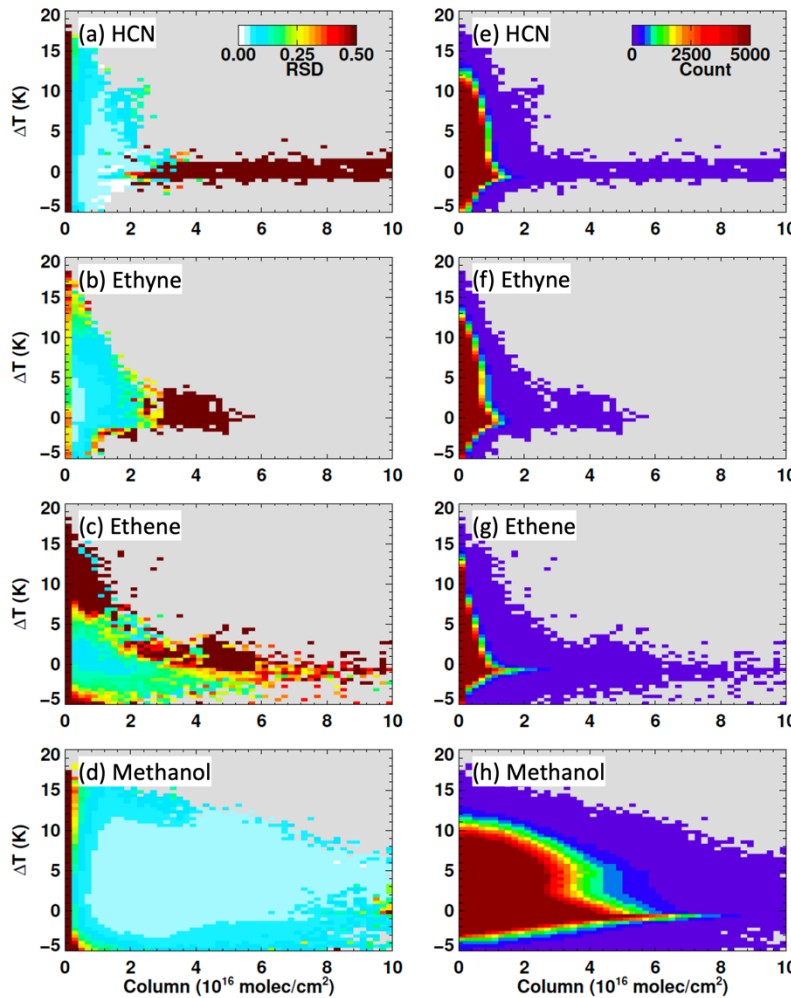

**Figure 5: ROCRv2 prediction precision as a function of column amount and thermal contrast.** Data plotted reflect the relative standard deviation of the 10-ANN predictions for each CrIS retrieval (left column) or the number of observations in each bin (right column) binned by thermal contrast (surface temperature – near surface air temperature) and column amount.





**Figure 6: Seasonal-mean HCN, ethyne, ethene, and methanol column enhancements as measured from CrIS using the ROCRv2 retrieval. Data shown are averaged across the SNPP-CrIS record (2012-2023) and represent above-background enhancements based on a globally-fixed profile shape ($P_{90}$ value) as described in-text.**





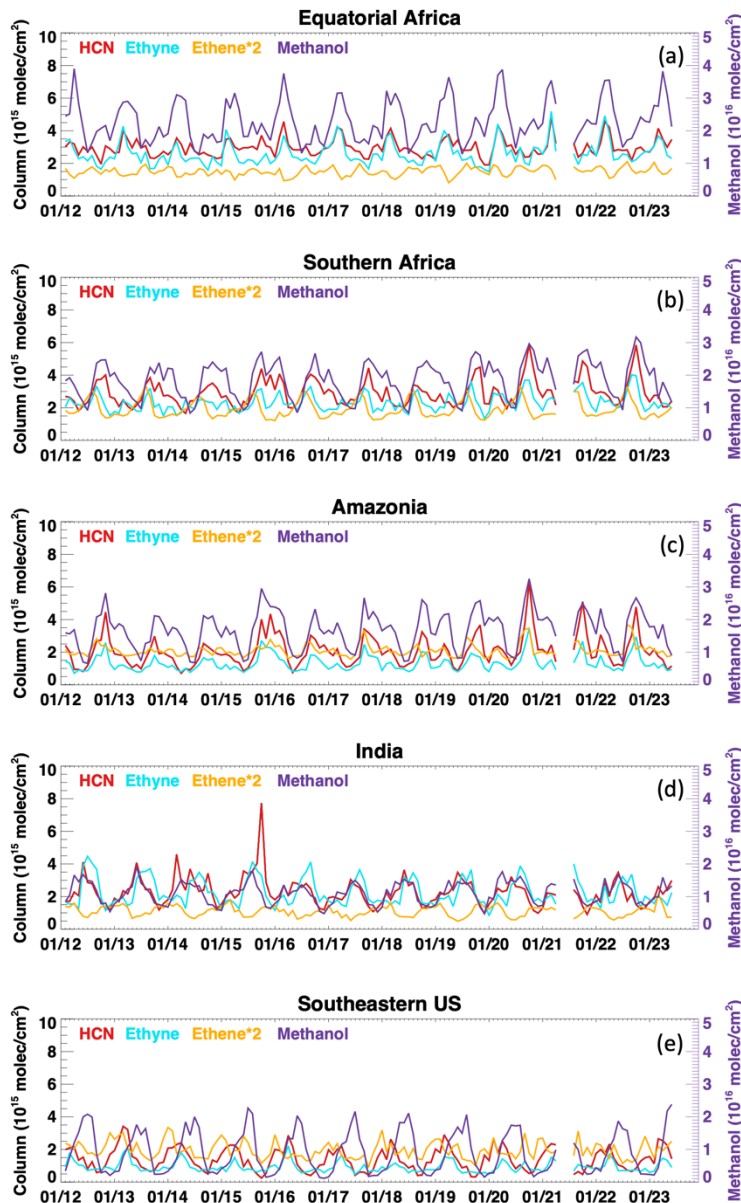

**Figure 7: Monthly mean VOC timeseries (2012-2023) as detected from SNPP-CrIS using the ROCRv2 retrieval. Data are shown for (a) equatorial Africa, (b) southern Africa, (c) Amazonia, (d) India, and (e) and the southeastern US, with regions mapped in Fig. S3. Quantities shown represent above-background column enhancements derived at a globally-fixed $P_{90}$ value as described in-text.**




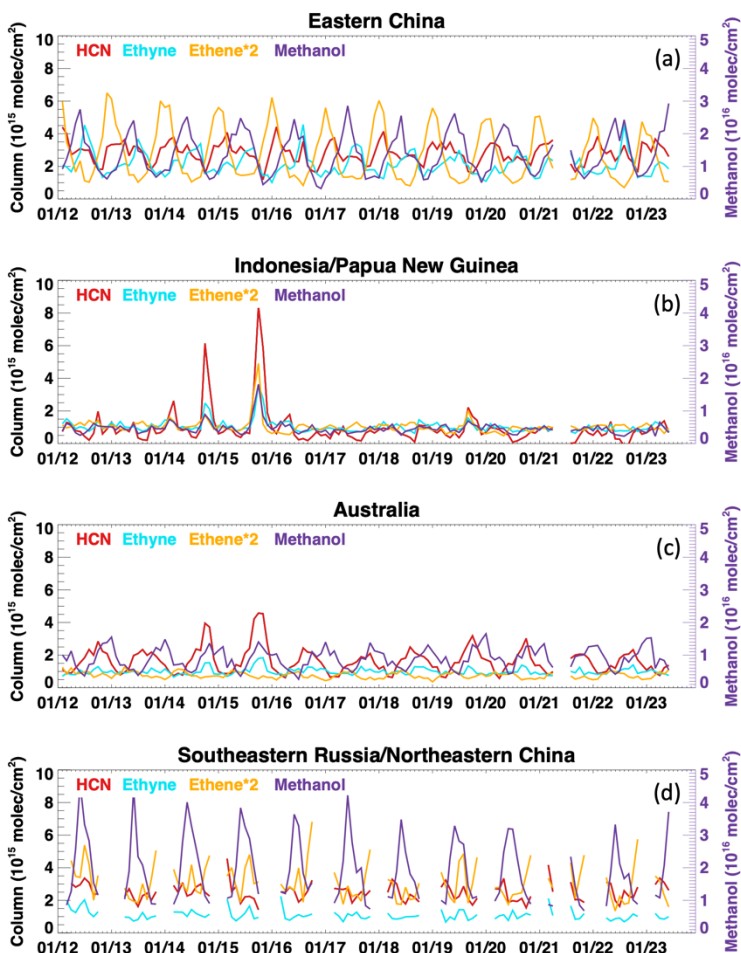

**Figure 8: Monthly mean VOC timeseries (2012-2023) as detected from SNPP-CrIS using the ROCRv2 retrieval. Data are shown for (a) eastern China, (b) Indonesia+Papua New Guinea, (c) Australia, and (d) and southeastern Russia+northeastern China, with regions mapped in Fig. S3. Quantities shown represent above-background column enhancements derived at a globally-fixed P$_{90}$ value as described in-text.**



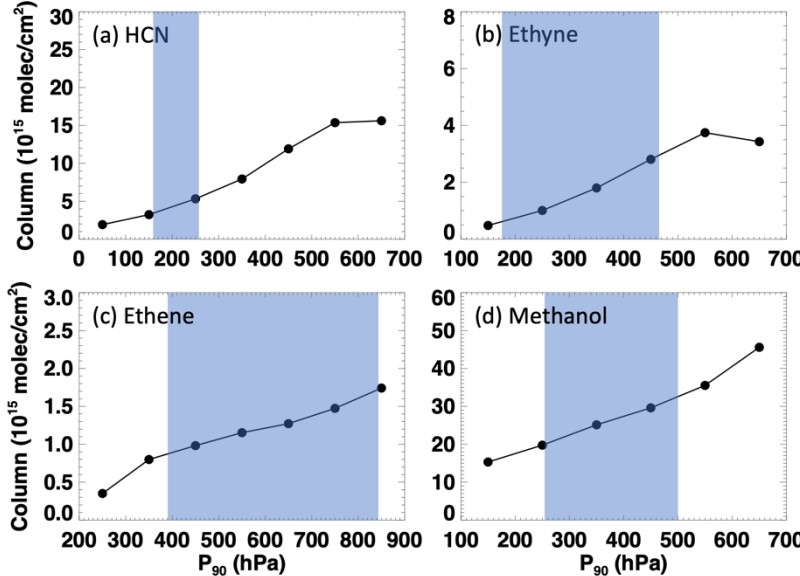

**Figure 9: Illustration of CrIS retrieval sensitivity to VOC vertical profile assumptions. Shown are the CrIS-retrieved column enhancements for October 2019 over Amazonia (mapped in Fig. S3) as derived using each of the globally-fixed $P_{90}$ values employed in the ROCRv2 ANN. Shading indicates the range in $P_{90}$ values encountered over this region based on GEOS-Chem predictions for each species.**



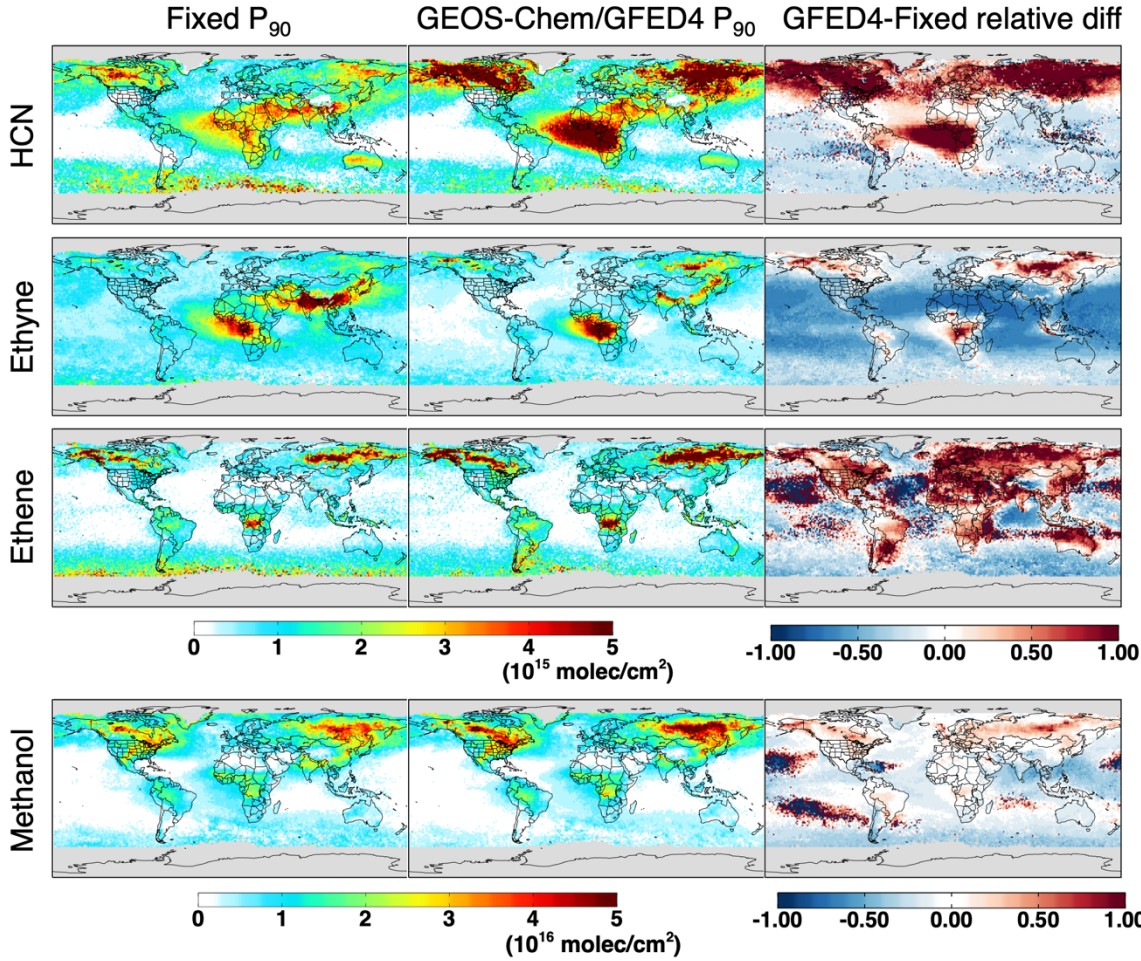

**Figure 10: Accounting for variable vertical VOC profiles in the CrIS retrievals. Plotted are CrIS-retrieved column enhancements as derived for July 2019 when using a globally fixed $P_{90}$ (left column), or when accounting for $P_{90}$ variability based on a GEOS-Chem simulation using GFEDv4 biomass burning emissions (middle column). Right column shows the relative difference between the two (GFEDv4-fixed $P_{90}$ column).**



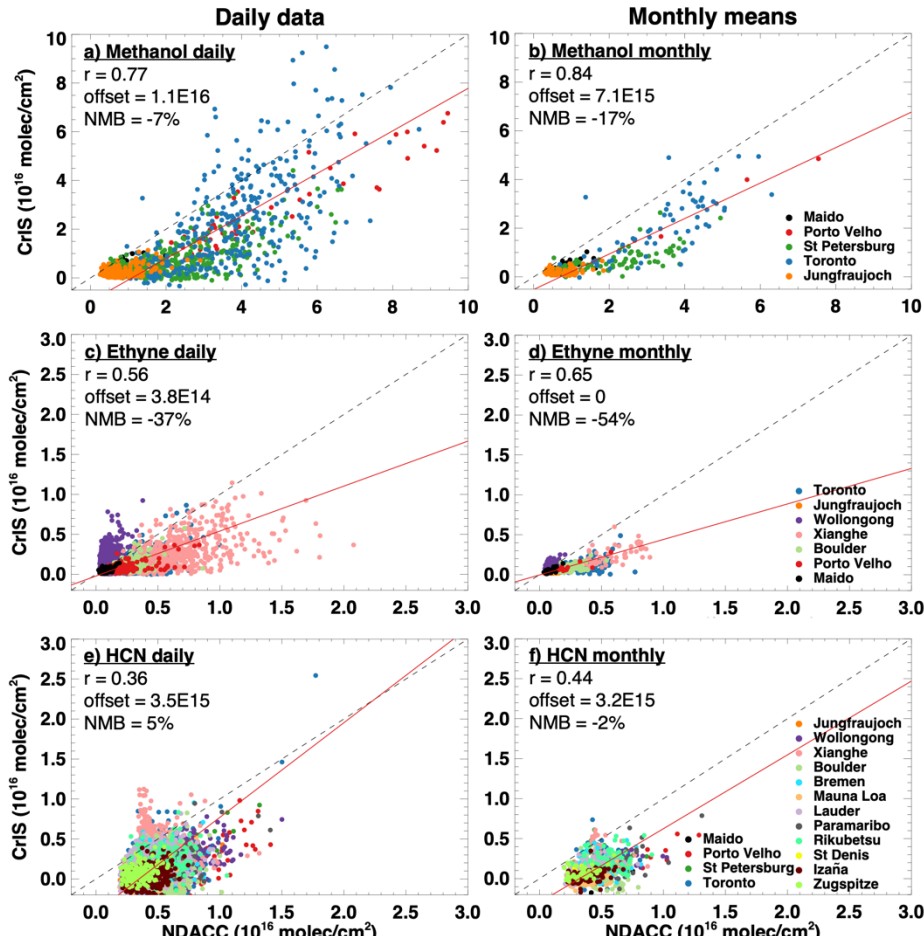

**Figure 11: Evaluation of the CrIS VOC retrievals against NDACC ground-based FTIR measurements. Data shown compare the CrIS column enhancements against the NDACC total columns on both a daily (left column) and monthly mean (right column) basis. CrIS retrievals have been interpolated to the NDACC-retrieved $P_{90}$ values and corrected for CrIS/NDACC elevation differences as described in the text. Normalized mean biases (NMB) are computed for the offset-corrected data.**



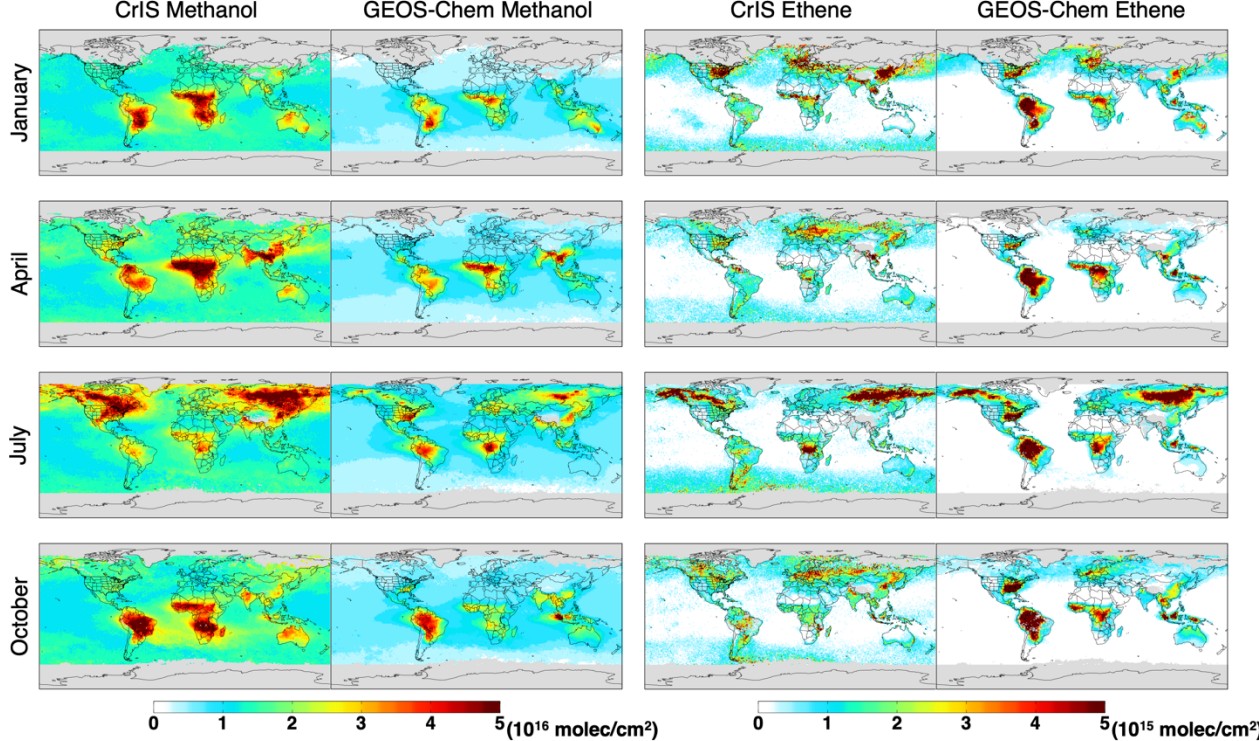

**Figure 12: Global methanol and ethene column distributions as observed by CrIS and simulated by GEOS-Chem for January, April, July, and October 2019. The CrIS data are adjusted to the model-predicted $P_{90}$ values in each case. Both datasets are screened as described in-text.**



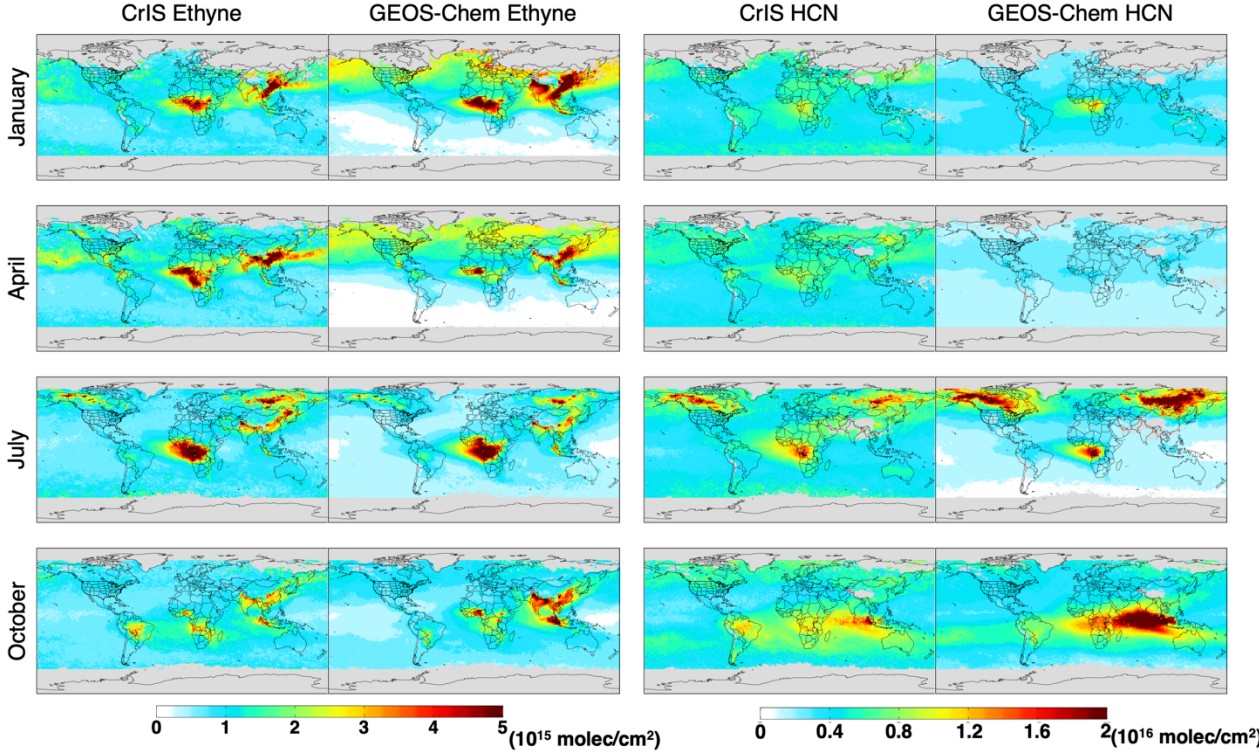

**Figure 13. Global ethyne and HCN column distributions as observed by CrIS and simulated by GEOS-Chem for January, April, July, and October 2019. The CrIS data are adjusted to the model-predicted P$_{90}$ values in each case. Both datasets are screened as described in-text.**

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
