# Peer review of "Global decadal measurements of methanol, ethene, ethyne, and HCN from the Cross-track Infrared Sounder"

_EGUsphere, 2024_

## Author Comment (AC1)

**Reviewer #1 response**

The manuscript presents global retrievals of four VOCs - methanol, ethyne, ethene, and HCN - from the polar-orbiting CrIS satellite sounder. First, their spectral signatures are identified in CrIS measurements taken in a fire plume from the 2019/2020 Australian bushfires. The retrieval methodology, which is based on a neural network approach, is then explained. Subsequently, the global distributions of the total columns of these VOCs are presented and discussed. CrIS data of methanol, ethyne, and HCN are compared with total columns derived from ground-based FTIR measurements at various NDACC sites, revealing notable correlations between spaceborne and ground-based measurements. Finally, the CrIS-derived distributions of these VOCs are compared with GEOS-Chem model simulations, and the discrepancies are analysed.

By presenting new satellite products of VOCs and conducting inter-comparisons with independent data and model simulations, the manuscript aligns perfectly with the scope of AMT. Although these four species have already been retrieved by other nadir-viewing or limb-sounding satellite instruments, these new CrIS products are of particular interest to the research field, given the increasing need for satellite observations of VOCs. With its low instrumental noise and early afternoon overpass time, CrIS shows promising capabilities for measuring VOCs. Overall, the paper is well-written and clearly structured, with results that are well-discussed and effectively supported by figures.

The retrieval framework is not new and builds upon the heritage of past work using other satellite instruments, which should be clearly recognized in the manuscript. The main new element is the introduction of a predictor for the vertical atmospheric distribution of VOCs, based on the relationship between the CrIS measured signal and GEOS-Chem vertical profiles. However, I have major concerns about this, detailed further below. The primary issue is that it might fall short of representing the full range of atmospheric variability of VOCs, as it constrains the CrIS retrievals within the range of VOC profiles from GEOS-Chem. As a global model, GEOS-Chem has a limited representation of the spatial and vertical variability of VOCs. For the same reasons, I do not think this predictor significantly improves the comparison with ground-based FTIR measurements, as would accounting for the a priori profile and vertical sensitivity respective to each instrument. I also have concerns about the feasibility of accurately retrieving these VOCs under all conditions, particularly at background concentrations, which might be within the instrumental noise. The example of VOC detections during the Australian bushfires presented in the manuscript is not convincing, as this event provided very favourable conditions. I have additional comments detailed below. I believe these comments need to be carefully addressed before proceeding further with the manuscript.

*We thank the reviewer for their careful read of the manuscript and their thoughtful and constructive feedback, which has improved this paper. Please find our responses to specific comments below in blue.*

**Major comments**

● Section 2.1: The cloud screening appears to be based on a simple temperature difference between the surface temperature and the brightness temperature from a single channel. The source of the surface temperature is unclear. If provided by CrIS, it is uncertain how the surface temperature can be obtained if the scene is cloudy. Additionally, when the surface temperature is low, clouds might have a similar temperature, making them difficult or impossible to detect. Using a single channel is also risky because it might not indicate residual clouds, while such clouds can affect the HRI and significantly bias the retrieved columns.

*The surface temperature is provided by MERRA-2 reanalysis data, which we now explicitly state in the paper. The reviewer is correct that the method can fail in cases of low clouds, particularly in marine environments when the cloud and surface temperature are similar. We have added new text to the paper mentioning this fact, and also pointing out that the majority of retained cloudy scenes occur over oceans where VOC concentrations are low. We have tested an alternative cloud screening based on the CrIS IMG product, which uses VIIRS imager information to derive cloud fractions on the CrIS footprint. However, this approach can erroneously remove smoke plumes, so we have opted to retain our current cloud screening approach as default and allow users to employ post-filtering using additional cloud metrics where appropriate. This point has also been added to the manuscript.*

● Lines 108-110 and Section 2: The entire retrieval methodology employed here, including the use of HRI and neural-network-based inversion, draws from approaches previously developed for the retrieval of ammonia and other species (including methanol) from IASI satellite observations, as described in published works (e.g., Whitburn et al., 2016; Franco et al., 2018). While it is standard practice to adopt already established methodologies, it is necessary to acknowledge this influence and explicitly reference the seminal studies.

*We agree and have already included two paragraphs in the introduction covering past work on TIR VOC measurements, upon which this work builds. We have now added a statement in the final introductory paragraph to make it clearer that our approach follows that used for IASI retrievals of ammonia and VOCs, citing the Whitburn et al. (2016) and Franco et al. (2018) studies (the former of which we inadvertently omitted from this paper).*

● Section 2.2: Detecting the spectral signatures of VOCs in CrIS radiances from a concentrated plume during the 2019/2020 Australian bushfires does not imply that these species can be detected elsewhere, in other conditions. These Australian fires represent by far the most favourable conditions in the last decade for detecting the spectral

signatures of any weak absorbers in satellite data (i.e., exceptional gas concentrations and very high injection height). The literature is full of such examples. Hence, this event cannot be used to demonstrate that VOCs can be detected and retrieved worldwide.

*As the reader notes, we have followed a standard practice of identifying the spectral signatures of our target species for a well-studied large event in which we can be reasonably confident they are present at high concentrations (and with significant vertical lofting). Our results demonstrate detectability in fire plumes, and also highlight the importance of our four target species in fire plumes (which in-situ measurements have shown to be among the four most abundant species emitted by fires; Permar et al., 2021, https://doi.org/10.1029/2020JD033838). We employ this as a first step and agree that it does not demonstrate detectability worldwide under all conditions, which is why we then proceed to validate our retrieval against other available measurements. We now make this more clear by pointing the reader here to the upcoming evaluation against NDACC in Section 4.*

● Following the previous comment, I have concerns regarding the global distributions of these VOCs, particularly in remote environments. What is the typical detection threshold of single CrIS pixel for these species? Can they truly be measured from individual CrIS spectra under all conditions? For instance, ethene columns below $1 \times 10^{15}$ molec cm$^{-2}$, derived from ground-based FTIR measurements at clean-air sites, are considered undetectable (Toon et al., 2018). The detection threshold with spaceborne instruments should be significantly higher. Therefore, it is important to consider whether robust retrievals are possible in remote areas, such as over the oceans, and what the satellite is actually measuring. Is it genuinely background abundance, or is it merely instrumental noise?

*By definition, the HRI-based retrieval quantifies tracer enhancements above background, so it cannot be used to characterize remote, background conditions. The precision of the ANN prediction (Fig. 4) provides one estimate of the level of detection. For our target species, the precision degrades (RSD > 60%) for column enhancements less than ~$2 \times 10^{15}$ molec cm$^{-2}$, and at somewhat higher columns for low to negative thermal contrast conditions. We've added some text in Section 2.3 noting this threshold as a level-of-detection, and reminding the reader that this represents an above background enhancement.*

● Lines 192-204: It is unclear why 10 different neural networks are trained, only to average their outputs to obtain a final prediction. Can the spread of predictions between these 10 networks truly be used to assess the performance of the training? If you use 50 or 100 different networks, the spread might be reduced, but this would be due to increased statistics, not necessarily because the training is better.

*The approach here is similar to cross-validation methods commonly applied in machine learning applications. The 10 networks are trained on 10 different extractions (i.e. subsets) of the synthetic data. The spread between the networks, defined as the relative standard deviation in our work, thus gives a measure of how robust they are to "unseen" data on which they were not trained. In fact the spread would not be reduced with the use of more networks. While this would occur if we were using a standard error metric (describing confidence in the mean), we are using the relative standard deviation (describing the distribution's spread). Therefore, if we trained more networks the same conclusions would hold. Wider inter-network spread can be used to identify more challenging retrieval scenarios; this information cannot be gained by training a network on a single subset (e.g. 50%, as used here) of the synthetic data.*

- The same remark applies to Lines 219-232: a filter is built based on the spread between the 10-ANN predictions to discard measurements with too low sensitivity. However, such sensitivity should depend primarily on the uncertainties of the input parameters (HRI, and other predictors), not on the spread between different networks. Indeed, the neural network is built with noise-free, synthetic data, while the retrievals are performed with actual inputs that have their own noise and uncertainties.

*The reviewer is incorrect in stating that the neural networks are built on noise-free data; they are built on synthetic data to which CrIS-like noise has been applied. We agree that input parameter uncertainties will contribute to the uncertainty in the final retrieved column values; in our view, evaluation against independent data (as performed in Section 4 of the paper) is the most robust way to assess such effects.*

*However, there are also environmental factors that control the sensitivity of TIR measurements independent of parametric uncertainties, and that is what we aim to account for in this analysis. Under low sensitivity conditions the retrieved columns are less robust and will thus vary more across neural networks.*

- Section 2.3: Input VOC profiles from GEOS-Chem are used to produce the dataset for the training of the neural network, and a P90 predictor, also derived from GEOS-Chem, is used to capture the variations in the spatial and temporal vertical distributions of the VOCs. However, the methodology for employing these profiles and P90 assumptions in the actual column retrievals is unclear. Is a single CrIS pixel retrieved multiple times for different P90 values? If P90 is available from external sources, is the gas column that is ultimately retrieved an interpolation between two columns previously retrieved using the two closest P90 assumptions? Clarification on this process is needed.

*Yes, the reviewer has correctly described the process. We have rearranged the paragraph that discusses this part of the method and also added some text to further clarify the methodology.*

- Throughout the study, the P90 is claimed to capture the vertical dependence of the VOCs and enable consistent vertical profile assumptions in comparisons between CrIS and models or other measurements. However, I disagree for several reasons:
  - Using VOC profiles and P90 values from GEOS-Chem constrains the CrIS retrievals within the range of variability of the model. This makes the retrievals dependent on a model, assuming that the model represents the true state of the VOCs. However, global models often fail to accurately represent the actual spatial and vertical variability of these species. This issue is exacerbated with fire plumes, which global models typically struggle to simulate accurately. Consequently, one can question how CrIS retrievals, trained and driven within the range of variability of the model, can produce more realistic representations of the VOCs.

*The reviewer is correct that global models can sometimes struggle to accurately capture the spatial and vertical variability of VOCs, especially in fire plumes. However, for the ANN training set, it does not matter whether individual plumes are represented correctly, as long as the model captures the dynamic range of the true state. Since we generate a set of retrievals at globally-fixed P90 values, a user can then interpolate based on observational constraints or P90 values from a particular model of interest.*

*Furthermore, it is a standard methodology to use modeled profiles as prior information in trace gas retrievals, so this issue is not unique to our study. In fact, work to date on machine-learning TIR VOC retrievals has typically used only a small set of vertical profiles (for example, IASI retrievals use two: one for land retrievals, and another for ocean; Whitburn et al., 2016; Franco et al., 2018) with no possibility for data users to adjust that assumption or characterize its effect. Our goal here was to provide a step forward by representing the vertical variation of VOC profiles in a parsimonious way that can be adjusted post-hoc by the user.*

*To the degree that the input model profile shapes do not encompass those in the real atmosphere, any out-of-sample conditions will have higher uncertainties as the reviewer suggests. We expect such issues to be most significant in fire plumes (with their potential lofting) and have added some text to Section 3.2 of the manuscript to this effect. Under other conditions, a large body of work has demonstrated that GEOS-Chem is able to capture a wide range of ambient VOC profile shapes (e.g., Xiao et al., 2007; Millet et al., 2008; Jin et al., 2023). It is the shape of these profiles, and not their magnitude or space-time accuracy, that matters in the context of our retrieval approach.*

○       While this approach facilitates comparisons between CrIS and GEOS-Chem, it does not enable internally consistent comparisons with other models, and even less so with other measurements. For a given P90 value, any vertical distributions can potentially occur in other models, and even more so with other independent measurements, which may significantly diverge from the vertical assumptions of GEOS-Chem. For example, in the CrIS-FTIR comparison, the range of FTIR retrieved profile shapes of VOCs below the P90 value likely differs significantly from those of GEOS-Chem. The neural network has not been trained to account for such profile variability, and the P90 value alone does not indicate the underlying profile.

*As discussed in our response to the previous comment, we facilitate comparisons to other models and independent data by providing the retrieval at multiple P90 values; the derived column can then be interpolated between these based on the local conditions. To the extent that the VOC profile shapes and P90 values in GEOS-Chem capture the dynamic range seen in the atmosphere and in other models, the approach enables internally consistent comparisons. Again, it is not the space-time accuracy of the profile shapes or their magnitude that matters here, but rather the fact that they span the range of true atmospheric conditions. The latter presumption is supported by the fact that GEOS-Chem, like all CTMs, uses assimilated, observationally-constrained meteorological fields to predict these profiles; it is also supported by the long history of VOC analyses using GEOS-Chem noted earlier. Injected fire plumes represent one particular challenge in this context, and we therefore include a sensitivity test using an alternative fire injection parameterization (GFAS) to illustrate the potential impact on retrieved columns. Data users can further evaluate such impacts based on their emission model of interest. Additional text addressing this caveat is now included in Section 3.2.*

○       Additionally, the P90 predictor does not account for the inhomogeneous vertical sensitivity of CrIS, which is likely degraded in the lower layers where the bulk of VOCs is typically found, affecting comparisons with models. Moreover, this sensitivity varies by instrument (the sensitivity of FTIR is supposed to be better).

*We disagree; the whole point of the P90 predictor is to characterize the inhomogeneous vertical sensitivity that the reviewer mentions. If the measurement sensitivity was vertically homogeneous there would be no P90 dependence. The P90 approach also accounts for horizontal variability in the CrIS vertical*

*sensitivity by quantifying the column-P90 relationship for each individual retrieval. Along with Figures 3 and 9, these points are also conveyed in Figure S6 of Brewer et al. 2024 and we now mention this in the paper to provide further context.*

*The reviewer is correct in stating that sensitivity varies by instrument, but our goal in this work is to allow data users to adjust the CrIS retrieved columns to any external constraint on the profile shape. This is a step forward over prior neural network-based approaches in which a vertical profile assumption is hard-wired with no way to change that assumption or characterize its effect.*

● A consequence of the CrIS retrieval dependency on GEOS-Chem can be observed in the limited extent of the training set in Figs 4 and 5. The number of synthetic predictions for high gas columns is low and quite limited to weak thermal contrasts, where, by definition, the sensitivity of the sounder is reduced. This might explain why, in Fig. 5, the prediction precision of HCN, ethyne, and ethene is poor for high columns, while this precision is better for lower columns with equivalent thermal contrast. The range of thermal contrast is also relatively narrow. It is not unusual to observe thermal contrasts beyond 15-20K. Therefore, it is questionable whether the neural network can generalize for observational conditions outside the conditions with which it has been trained.

*Yes, neural networks cannot generalize well for conditions outside of which they have been trained; this is a universal issue in machine learning. Our training set is based on a best estimate of real atmospheric conditions (as constrained by assimilated meteorology) and expanded via 100% randomization. Rare events, such as the high column, low thermal contrast episodes discussed at Line 227, do occur that by definition are not well-represented in our training set. As for thermal contrast, the combination of very high (>15-20 K) thermal contrast values with elevated VOC columns is also quite rare and thus represents a challenging scenario for the ANN. We are currently working on retrieval refinements for fire plumes, which will involve training sets with an expanded representation of anomalous events.*

*We have modified the text to further emphasize the point that we do not necessarily expect the ANN to perform well under anomalous conditions that are not statistically well-represented in the training.*

● Section 4: This section lacks a comprehensive explanation of how the FTIR-CrIS comparison is performed. In particular, are the FTIR and CrIS measurements co-located in space and time? It is mentioned that CrIS retrievals are interpolated to the local P90 value from the FTIR retrieved profile. Are the FTIR and CrIS measurements compared in pairs? However, the comparisons are shown for daily and monthly averages. For days

where FTIR measurements are available for part of the day (e.g., in the early morning only), is the subsequent daily average really comparable with the CrIS overpass, given the variability of species like methanol and ethene?

*Yes, the FTIR and CrIS measurements are collocated in space (taken as the closest CrIS gridbox to the FTIR measurement) and time (where daily means of FTIR data are compared to each CrIS measurement), and are compared pair-wise. We tested a stricter time match criterion (+/- 2 hours) but it had little impact on the comparison. The monthly means represent the average of all days in the month in which both instruments provided a measurement. We have updated the text to clarify this.*

**Minor comments / typos**

● Title: Typically, "long-term" refers to climatological series spanning at least 20-30 years. Perhaps "decadal" would be more appropriate.

*We have changed "Long-term global"to "Global decadal" in the title.*

● Lines 95-98: There is also a more recent, neural-network-based HCN product from IASI published by Rosanka et al. (2021).

*We appreciate the catch here and have added this citation.*

● Lines 155-158: The 2019/2020 Australian bushfires are known for their heavy smoke aerosol content. Are these aerosols accounted for during the fit? Can the residuals of an in-plume spectrum, likely affected by broadband absorptions from smoke aerosols, be compared with those from an out-of-plume spectrum that is free of such aerosols?

*Given the small size of the biomass burning aerosols relative to the long wavelengths being examined here, we don't expect the residuals to be affected by aerosol absorption in these events.*

● Lines 161-162: "confirming their importance in fire plumes and the underlying spectral signals driving the HRI values." The spectral fits in Section 2.2 are performed to demonstrate that the spectral signatures of the VOCs are present in CrIS radiances. However, from what I understood, the HRI is not a fit but is calculated over an entire spectral range that is larger than just the spectral features highlighted by the residuals of the fits. Can it be confirmed that the signal contributing to the HRI comes solely from the targeted VOC species and that no other absorbers contribute significantly?

*Yes, the spectral fits are done to confirm the spectral signatures of our target species in the CrIS radiances, which in turn confirms the presence of each species itself in the plume. The test does not necessarily confirm that other absorbers are not contributing to the CrIS HRIs for our target species, which is why this and other retrieval approaches must always be evaluated against independent data. However, as noted in the paper, IASI analysis of this same event using a different approach (spectral whitening, De Longueville et al., 2021) also demonstrates the specific contribution of these same compounds which provides further confidence in the CrIS attribution.*

● Lines 190-191: Ozone is a major interference in the absorption band of methanol, and HCN and ethyne (and even ethene) absorb in a spectral range with significant absorption features of CO2. Are these potential interferences accounted for in the retrieval? If not, what could be the impact on the retrieved columns? In addition, the HRI is set up based on CrIS observations from 2019, but then applied to the entire 2012-2023 time series. During this time span, CO2 concentrations have increased from <400 ppm to >420 ppm. Could this bias the retrieved VOC columns before and after 2019?

*Ozone is a potential interferent for methanol; however, we found minimal impact on the ANN training performance when we included it as a predictor in the ANN.*

*We tested the impact of CO2 on our results by scaling the CO2 amounts used in our 2019 simulation. We found that an 8% increase in the CO2 concentrations led to changes in the simulated HRI of ~5% for HCN and ~3% for ethyne (with a near-zero impact for ethene and methanol). As the reviewer notes, the actual CO2 change across the S-NPP CrIS record is on the order of 5%. Therefore, we expect minimal bias associated with CO2 changes on a decadal timeframe, but will explore the need for a time-specific CO2 attribution as we build longer VOC records from the CrIS sensors.*

● Lines 190-191: Are three levels enough to describe the atmospheric temperature profile for the retrieval? There might be significant variability between these levels. Additionally, in Lines 210-211, what do you mean by "dispersed among multiple input variables that individually have only minor impacts"? What are these variables? An accurate representation of the temperature profile is usually important, and misrepresentation can lead to biased retrieved concentrations.

We include 5 temperatures in our retrieval: the surface skin temperature, surface air temperature, and temperatures at ~850, 625, and 225 hPa. We did not include additional temperature variables to avoid NN overfitting to non-HRI variables. The text "multiple input variables that individually have only minor impacts" is referring to each of the 5 temperatures, not any additional variables. We have updated the text to make this clearer.

● Lines 206-207: The vertical location of the VOC in the atmosphere influences the thermal contrast. Isn't this the same as saying that the measured signal is largely dependent on the temperature profile and surface temperature?

*The reviewer here is referring to the variable withholding tests, in which we note that the HRI is the dominant predictor followed by the P90. Yes, these are two ways of saying the same thing: for a given thermal state, measurement sensitivity depends on the vertical location of the VOC. Equivalently: for a given VOC profile, measurement sensitivity depends on the thermal structure of the atmosphere. We have added text to this section noting that the P90 dependence reflects the underlying effect of surface-absorber thermal contrast.*

● Line 209, typo: "yield"

*We feel the current form, "yields", is correct to get proper subject-verb agreement (i.e., "Withholding…yields…")*

● Lines 171-177: I don't understand why each simulation is replicated 25 times with random noise, only to take the mean of all 25 replications to obtain the final HRI value. This process cancels out the noise that was initially added.

*Because the CrIS HRI values are aggregated to the MERRA2 grid prior to performing the retrieval, the 25 replications (and subsequent averaging) are used to simulate aggregation in the ANN network. We have added some text to clarify this. Because the noise is applied to the radiances prior to the HRI calculation (rather than being added to the HRI itself), it does not cancel out across the 25 replications.*

● Figure 6: What is responsible for the high HCN and ethene columns observed in winter over the North Atlantic Ocean and along the eastern coast of Siberia? It seems unlikely that these are due to outflows from continents. Additionally, concerning ethyne, we do not observe strong latitudinal gradients between the Northern and Southern Hemispheres, which contrasts with what might be expected for an anthropogenic hydrocarbon.

*We discussed the elevated wintertime HCN and ethene columns over high latitude oceans in lines 259-261 of the paper. We also see enhancements over the southern ocean winter (JJA, discussed in line 285-286), so we suspect these arise from surface interferences in the presence of sea ice or whitecaps. However, future work is needed to confirm this.*

*As for the lack of a latitudinal gradient in the CrIS ethyne, part of this may be due to the fact that a globally-fixed P90 is applied in Fig. 6. A spatially varying (model-based) P90 is applied in Fig. 13, and the result is a more pronounced latitudinal gradient in the northern hemisphere winter. However, the CrIS gradient is still weaker than what is predicted by GEOS-Chem, which may in part reflect higher retrieval uncertainty in near-background conditions.*

● Figure 7: The seasonal cycle of ethene in India contrasts sharply with that of the other species. Why?

*Future work is needed to investigate the drivers of the observed seasonality here and elsewhere, but the wintertime peak in ethene over India suggests an anthropogenic source whereas biogenic and pyrogenic sources drive summertime peaks for the other species. A similar contrast is observed over East Asia, where large signals detected by IASI have been linked to industrial sources (Franco et al., 2022).*

● Figure 11: The daily CrIS data show negative columns of methanol and HCN in the scatter plots, but this is not the case with the CrIS ethyne data (Fig. 11c). Were those filtered out?

*Negative columns do also exist in the ethyne data and have not been filtered out in Fig 11, they are just small in magnitude compared to the positive values and thus difficult to see on the scale used.*

● Section 4: CrIS column offsets are derived from the comparison with FTIR columns, based on the intercept of the linear regression between the two datasets. However, there is considerable disparity around this regression, which seems to be partly driven by the FTIR sites included in the comparison. How do these intercepts, and hence the column offsets, vary if more or fewer FTIR sites are included in the comparison?

*We now include a table in the SI with the correlation coefficient and offset for the CrIS:NDACC comparison at each individual site. Correlation coefficients and residual biases for individual site monthly means were already included in the timelines in Figs S8-S11.*

● Line 311: "VOC spatial distributions". Do you mean "VOC vertical distributions"?

*Yes, we appreciate the catch and have made this edit.*

● Lines 330-333: Few measurements of ethene columns exist, but total columns have been retrieved in various environments up until 2016 with the MkIV FTIR instrument (Toon et al., 2018), including urban sites with high ethene concentrations. Additional ground-based FTIR retrievals of ethene were conducted at the sub-polar Eureka NDACC site in Canada during the overpass of a fire plume (Wizenberg et al., 2023). These measurements could be valuable for comparison with CrIS data.

*We thank the reader for noting this. We have done some analysis of the MkIV FTIR ethene– most of the deployments occurred prior to the CrIS launch, but we do have a comparison at the JPL site. We now include a plot of the comparison at this site in SI, which yields a moderate correlation (r=0.44). The CrIS radiances are not processed at high latitudes (>~70 degrees), so we do not have retrievals to compare to measurements at the Eureka site.*

● Lines 376-380: I believe that the DOFS is not a valid argument to explain the weaker CrIS/NDACC agreement for HCN and ethyne. The DOFS values for methanol, HCN, and ethyne retrieved from ground-based FTIR measurements typically range between 1 and 2, rarely exceeding 2. Therefore, the DOFS values are similar for these three species at most FTIR sites. Instead, I lean towards the view that the weaker CrIS/NDACC agreement for HCN and ethyne may be attributed to the GEOS-Chem-driven training of CrIS retrievals, which may be less robust in capturing the variability seen in retrieved FTIR profiles for these two species.

*In this section we did not intend to claim that the NDACC DOFS were the reason for the weaker NDACC/CrIS agreement. Rather, we are hypothesizing that uncertainties in the vertical profile shapes, and their effects on the CrIS retrievals, are a contributing factor, particularly for HCN and ethyne. In this section we have used the retrieved NDACC profiles to derive P90 values that are then applied to the CrIS retrievals. We mention the NDACC DOFS because (as the reviewer points out) the NDACC measurements of our target species do not typically have sufficient vertical information to fully characterize the profile shape. We have edited the text to clarify these points.*

---

## Author Comment (AC2)

**Reviewer #2 response**

*We thank the reviewer for their thoughtful comments, which have improved this paper. Please find our responses below in blue.*

- Line 185-187. This line needs more evidence, as without an averaging kernel this retrieval seems to be too heavily reliant on strong vertical mixing to be able to detect these species. These species are likely not most sensitive at the surface so any location with a high P90 value is probably missing a significant amount of information.

  *In these lines we are simply stating that the P90 gives us a compact (i.e. succinct) way to represent the dependence of the HRI-column relationship on the height of the absorber in our retrieval framework. The reviewer is correct that the sensitivity will be decreased when the P90 value is high (i.e. closer to the surface), but this is reflected in the lower HRI values for a given column amount under these conditions. We have changed the word "captures" to "accounts for" in case the former word overstates our ability to remove uncertainties associated with vertically-dependent sensitivity and mixing.*

- Following the previous comment, it would be helpful to explain how the P90 is applied from the GEOS-Chem grid to match the gridded HRI values. This would likely create problems in regions with strong elevation gradients or strong emissions sources.

  *The GEOS-Chem simulations employed in our analysis are at 2° × 2.5° resolution, so we use the closest gridbox to match the P90 with the corresponding gridded CrIS HRI. We have added some text to Section 3.2 to clarify this. We agree with the reviewer that using a P90 from a coarser simulation will lead to some errors in the vicinity of strong gradients and/or localized sources. We have added text to Section 5 to note that this is something to consider in our model-measurement comparisons, and note that higher-resolution simulations would yield finer-scale information in future studies.*

- Section 4 could explain more how the NDACC measurements are synced to the CrIS overpass time. Also, are the concentrations being compared via the 1 grid cell that contains each site? If so is there any meteorological screening applied to avoid abnormal conditions? Needs to be explained more thoroughly.

  *Yes, the FTIR data are compared to the CrIS data for the gridbox containing the FTIR site. We have updated the text to note this. We do not apply any meteorological screening to the comparison, but neither dataset includes measurements under cloudy conditions.*

---

## Author Comment (AC3)

**Community comment #1 response**

*We thank the Dr. Solomon for sharing their questions on our paper. Please see our responses below in blue.*

1) ENSO: while the results for 2016 are interesting, is anything seen in the last year's data? To be convincing regarding ENSO signals, it would be helpful to discuss additional information about the year 2023 through the end of this very hot year

   *This paper only presents analysis from CrIS onboard the Suomi-NPP satellite; that instrument lost its longwave channels in August 2023, so our record for that year does not extend past July, unfortunately. Our understanding is that the Indonesian fire season peaks in October, so we are unable to see signals of ENSO-induced drought in that last year. We have seen a number of news stories pointing to 2023 as a big fire year in Indonesia, and we look forward to analyzing those events further with the NOAA-20 and NOAA-21 CrIS instruments.*

2) I am very surprised not to see enhanced HCN or other wildfire markers over Australia during the remarkable 2020 wildfire season in Figure 8. Can the authors please explain what they think about the data in that year in more detail?

   *While the 2020 fire season does exhibit the largest peak in methanol over Australia, because these plots are monthly means averaged over the whole continent, we don't see large signals associated with that event in these timeseries. The impact of those fires would likely be more apparent had we included downwind ocean regions in these timeseries, since we do see strong signals in the lofted 2020 Australian fire plumes as discussed in Section 2.2.*

3) It would also be helpful to see time series plots similar to figures 7 and 8 for Siberia, where certain years displayed high frequency of fires, and for Canada. One would imagine that this instrument would have seen a strong wildfire signal in 2023 over Canada. Does it?

   *Because of the Suomi-NPP CrIS record ends in summer 2023, as mentioned above, the full 2023 Canadian fire season is not represented in our dataset. However, we look forward to investigating this further with the NOAA-20 and NOAA-21 instruments.*

---

## Author Response (AR2)

We sincerely thank the reviewers and the editor for their feedback and support of our revised manuscript. Per the editor's request, we have uploaded the data and code associated with this paper to the University of Minnesota Digital Repository. It is currently being reviewed by their curators and will soon be assigned a doi. We have put a placeholder doi (https://doi.org/XXXX) in the 'Code availability' and 'Data availability' sections of the paper, and will replace this with the actual doi when we receive the proofs.